# Dataset Distillation for Pre-Trained Self-Supervised Vision Models

**George Cazenavette**  **Antonio Torralba**  **Vincent Sitzmann**

Massachusetts Institute of Technology

georgecazenavette.github.io/linear-gm

## Abstract

The task of *dataset distillation* aims to find a small set of synthetic images such that training a model on them reproduces the performance of the same model trained on a much larger dataset of real samples. Existing distillation methods focus on synthesizing datasets that enable training *randomly initialized* models. In contrast, state-of-the-art vision approaches are increasingly building on large, pre-trained self-supervised models rather than training from scratch. In this paper, we investigate the problem of distilling datasets that enable us to optimally train *linear probes* on top of such large, pre-trained vision models. We introduce a method of dataset distillation for this task called *Linear Gradient Matching* that optimizes the synthetic images such that, when passed through a pre-trained feature extractor, they induce gradients in the linear classifier similar to those produced by the real data. Our method yields synthetic data that outperform all real-image baselines and, remarkably, generalize across pre-trained vision models, enabling us, for instance, to train a linear CLIP probe that performs competitively using a dataset distilled via a DINO backbone. Further, we show that our distilled datasets are exceptionally effective for fine-grained classification and provide a valuable tool for model interpretability, predicting, among other things, how similar two models' embedding spaces are under the platonic representation hypothesis or whether a model is sensitive to spurious correlations in adversarial datasets.

## 1 Introduction

The task of *Dataset Distillation* involves the synthesis of a small set of synthetic samples such that a model trained *from scratch* on this synthetic set will achieve test-time performance comparable to that of a model trained on the full real dataset. Since this problem's first introduction and proposed solution in the self-titled paper [47], many new methods [6, 27, 51, 54–56] and extensions thereof [7, 11, 16, 25, 28, 40, 46, 53] have made strides towards the lofty goal of learning a high-quality model from just a handful of synthetic images.

Meanwhile, computer vision has increasingly adopted a paradigm of using the representations of large, pre-trained self-supervised vision models for downstream tasks, either via fine-tuning or by using these models as feature extraction backbones. Given this trend, in this work, we explore dataset distillation in the regime of training models *on top of* features extracted by pre-trained vision foundation models. Specifically, we study **linear classification** on top of a **pre-trained feature representation**.

In our new method, *Linear Gradient Matching*, we distill synthetic datasets by optimizing such that their representations extracted by pre-trained feature extractors induce *gradients* in a linear classifier similar to those obtained from real images. We find that a single synthetic image per class suffices to train linear classifiers to competitive performance across a wide variety of large vision

39th Conference on Neural Information Processing Systems (NeurIPS 2025).

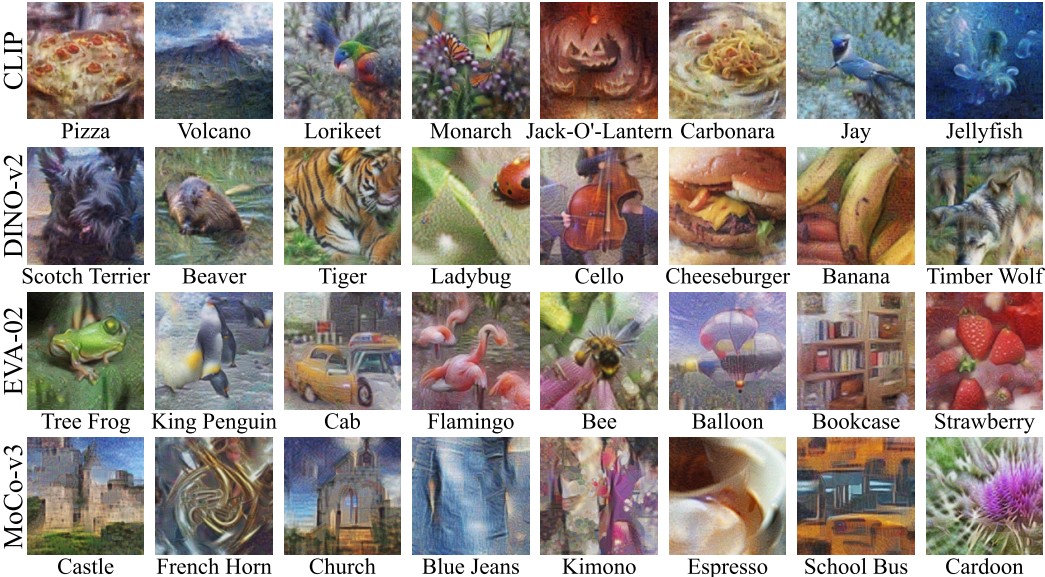

Figure 1: **ImageNet-1k Distilled for Self-Supervised Models:** Using our method of *Linear Gradient Matching*, we distill vision datasets to just one synthetic image per class using different pre-trained self-supervised backbone models. These learned images can then be used to train linear probes that achieve high accuracy on unseen test data, outperforming all real-image baselines. Furthermore, each backbone model seems to yield its own "style" of distilled image, giving insights into the aspects on which these models tend to focus (structure, texture, color, etc.).

model backbones, outperforming all real-image baselines. Figure 1 shows samples distilled from ImageNet-1k [12] with our method using various self-supervised feature extractors.

Motivated by recent hypotheses that different large models converge to similar representations even when trained on different modalities [20], we investigate whether distilled datasets transfer *across architectures*. We find that a gradient matching objective alone leads to images that are overfit to a particular model architecture and do not yield competitive performance across foundation models. However, we overcome this issue through differentiable augmentations and a simple re-parameterization of images via a multi-scale pyramid. Compared to those retrieved via naïve pixel optimization, the resulting distilled images not only look remarkably realistic but also readily transfer across foundation models, such that a dataset distilled using, for example, a DINO backbone yields competitive performance when used to train a linear classifier on top of a different model's representation, such as CLIP's.

We also observe that our distilled datasets offer several interesting interpretability results, including predicting alignment between different models, explaining susceptibility (or robustness) to spurious correlations in adversarial datasets, and highlighting out-of-distribution capabilities.

Extensive experiments and ablations validate our *Linear Gradient Matching* method's effectiveness on this new dataset distillation task and highlight its potential as an interpretability tool.

## 2   Related Work

**Dataset Distillation.**   As dataset and model sizes continue to grow, so has the interest in more efficient forms of learning. To this end, researchers have worked towards methods of *learning* optimal training data such that one could train an effective model from scratch using as few samples as possible. One such solution to this problem was the initial proposal of *Dataset Distillation* [47] in which the model's final performance was expressed as a function of the *synthetic training data* that was optimized end-to-end by back-propagating through many inner training iterations. Follow-up works introduced proxy losses and learned synthetic images that matched gradients [55], feature distributions [54], training trajectories [6] and more [27, 51, 56]. Some works extend dataset distillation to large models [50, 51], but these methods do not excel in the ultra-small data regime, i.e., one image per class. For such settings, Trajectory Matching [6] and

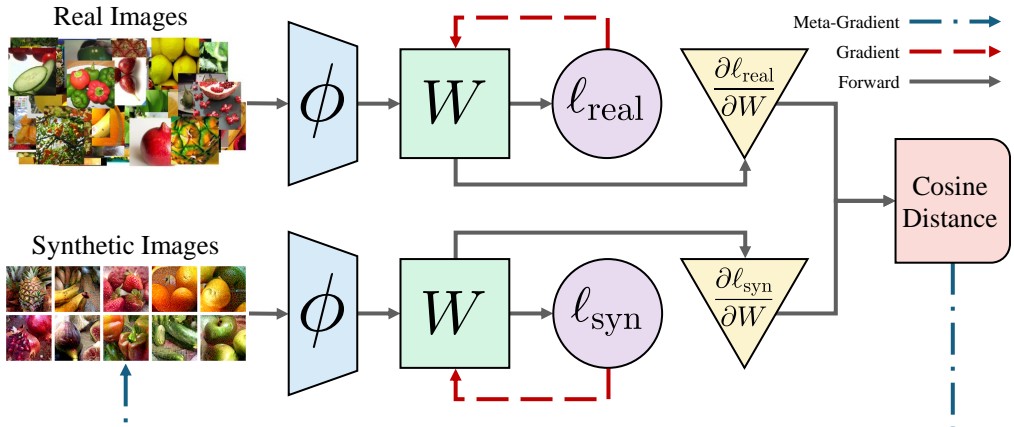

Figure 2: **Linear Gradient Matching for Pre-Trained Vision Models:** Given a pre-trained self-supervised vision model ($\phi$), we perform a distillation step by first passing a batch of real and synthetic data through $\phi$ and a randomly-initialized linear classifier ($W$) to get the real and synthetic classification losses ($\ell_{\text{real}}$ and $\ell_{\text{syn}}$). Our meta loss ($\mathcal{L}_{\text{meta}}$) is then defined as the cosine distance between the *gradients* of these classification losses ($\ell_{\text{real}}$ and $\ell_{\text{syn}}$) with respect to the random linear probe ($W$). This meta loss is then back-propagated through the initial synthetic gradient calculation and used to update our synthetic images. This technique allows us to distill large datasets to just a single image per class while still achieving high performance when training new linear probes.

its modifications [7, 11, 16, 32, 40] still reign supreme. However, this method fails to scale up to large models due to memory constraints and instability in the bi-level optimization.

This work introduces a new problem in the space of dataset distillation: learning synthetic images for the purpose of training *linear probes* on top of *pre-trained self-supervised feature extractors* instead of training randomly initialized models from scratch. Our proposed solution, *Linear Gradient Matching*, takes inspiration from prior work on gradient matching [55] and trajectory matching [6] but only considers gradients of the linear classifier as opposed to the entire model.

**Self-Supervised Learning.**    Given that the overwhelming majority of available visual data lacks any useful labels, *Self-Supervised Learning* has become the defacto method of pre-training neural networks to be later used for down-stream tasks. In recent years, several different paradigms of self-supervised training have emerged, including contrastive learners [5, 8–10, 18, 31], masked auto-encoders [17, 37], vision-language models [34, 41, 44, 52], and hybrid approaches [13, 14]. Despite the various training methods, researchers have noticed that these different models tend to learn similar representations, even across different modalities, and dubbed this observation the "Platonic Representation Hypothesis" [20].

In this work, we focus on distilling datasets using four pre-trained self-supervised models in particular: CLIP [34], DINO-v2 [31], EVA-02 [13], and MoCo-v3 [10]. Since our feature extractors were pre-trained in a purely self-supervised manner, our linear probes still only see a single *labeled* sample per class while achieving competitive performance.

## 3   Method

*Dataset Distillation* aims to synthesize a tiny synthetic dataset of (typically) images that are optimal for training. Unlike previous works designed to produce synthetic samples for training new models *from scratch*, our new method, *Linear Gradient Matching*, aims to distill datasets for the purpose of training **linear classifiers** in the embedding space of **pre-trained feature extractors**.

### 3.1   Linear Gradient Matching

Formally, given a pre-trained self-supervised feature extractor, $\phi$, along with a real dataset $D_{\text{real}}$ of images $X_{\text{real}}$ and labels $Y_{\text{real}}$, we wish to distill a small synthetic set $D_{\text{syn}} = \{X_{\text{syn}}, Y_{\text{syn}}\}$ such that $D_{\text{syn}}$ can be used to train a linear classifier in feature extractor $\phi$'s embedding space that will have similar test-time performance to one trained on $D_{\text{real}}$.

Inspired by previous works that aimed to match single-step gradients [55] or multi-step trajectories [6], our solution is designed to ensure that training on our synthetic dataset results in similar updates as training on real data. In other words, the gradients of the classification loss using synthetic images (with respect to the linear classifier) should match those using real images.

To achieve this, we sample a random linear classifier matrix $W \sim \mathcal{N}(0,1)^{c \times f}$ at each distillation step where $c$ is the number of classes and $f$ is the feature extractor $\phi$'s output dimension. After passing the real and synthetic images through the feature extractor $\phi$ and linear classifier $W$, we find the real and synthetic classification losses where CE is the multiclass cross-entropy loss:

$$\begin{aligned} \ell_{\text{real}} &= \text{CE}(W\phi(X_{\text{real}}); Y_{\text{real}}) \\ \ell_{\text{syn}} &= \text{CE}(W\phi(X_{\text{syn}}); Y_{\text{syn}}) \end{aligned} \tag{1}$$

We then take the gradients of these classification losses with respect to the linear classifier $W$, and our *meta loss* is then the cosine distance between them:

$$\mathcal{L}_{\text{meta}} = 1 - \cos\left(\text{vec}\left(\frac{\partial \ell_{\text{real}}}{\partial W}\right), \text{vec}\left(\frac{\partial \ell_{\text{syn}}}{\partial W}\right)\right) \tag{2}$$

This meta loss is then back-propagated through the inner gradient computation, linear classifier, and feature extractor to find $\partial \mathcal{L}_{\text{meta}} / \partial X_{\text{syn}}$ and update our synthetic images. This process is repeated until distillation is complete. An outline of this *Linear Gradient Matching* method can be found in Figure 2.

## 3.2 Implicit Regularization

As noted in prior works [6, 7, 55], *Dataset Distillation* tends to yield synthetic images that are *overfit* to the model used to perform the distillation. This issue manifests in what appear like adversarial patterns and inhibit the images' usefulness when used to train other models.

One recent work [15] proposed using a pyramid representation rather than naïve pixels for image optimization problems. While the authors used this technique for CLIP inversion, we find it works remarkably well for our Dataset Distillation task. Rather than simply optimizing pixels, each synthetic sample is instead stored as a set of images of different resolutions $\rho = \{1 \times 1, 2 \times 2, 4 \times 4, \ldots, 256 \times 256\}$.

Before each optimization step, the composite images $X$ are "rendered" by bilinearly upsampling each level of the pyramids $P = \{P_r | r \in \rho\}$ to the max resolution (256) and adding them together before smoothly clamping the pixel values with a sigmoid function:

$$X = \text{sigmoid}\left(\sum_{r \in \rho} \text{resize}_{256}(P_r)\right) \tag{3}$$

Furthermore, we progressively optimize our pyramid, starting with just the lowest-resolution component and periodically adding more tiers during distillation. The effects of using the pyramid representation are quite dramatic, as seen in Figure 5.

As an additional step to combat overfitting, we also learn our distilled images in a *decorrelated* color space, as described in prior feature visualization work [30]. In short, we apply a fixed linear transform to the channels of our images after the pyramid reconstruction that brings them back into the standard correlated color space. This helps ward off any potential color-based biases induced by the model used during distillation.

## 3.3 Differentiable Augmentations

As first noted in the work on *Differentiable Siamese Augmentation* [53], applying differentiable augmentations to the synthetic images during the distillation process greatly improves the quality of the distilled data. As such, we apply **horizontal flipping**, **random resized cropping**, and **Gaussian noising** to our distilled images at each step. In practice, we actually apply multiple rounds of this augmentation to different copies of the synthetic data at each iteration and concatenate the results. We find this improves distillation since the optimization now encourages all these augmented copies *together* to be the ideal training set rather than attempting to fit all pertinent information into a single augmented version of the images.

| Train Set (1 Img/Cls) | ImageNet-100 | | | | | ImageNet-1k | | | | |
|---|---|---|---|---|---|---|---|---|---|---|
| | CLIP | DINO-v2 | EVA-02 | MoCo-v3 | Average | CLIP | DINO-v2 | EVA-02 | MoCo-v3 | Average |
| Distilled (Ours) | **84.9**±0.1 | **91.5**±0.1 | **89.0**±0.0 | **83.4**±0.1 | **87.2**±0.1 | **63.0**±0.0 | **75.0**±0.1 | **70.3**±0.1 | **63.2**±0.0 | **67.9**±0.0 |
| Neighbors | 67.8±0.3 | 86.0±0.2 | 78.8±0.2 | 77.1±0.1 | 77.4±0.2 | 38.8±0.1 | 67.7±0.1 | 49.9±0.1 | 56.4±0.0 | 53.2±0.1 |
| Centroids | 77.1±0.1 | 86.9±0.3 | 80.9±0.2 | 77.7±0.1 | 80.6±0.2 | 53.9±0.0 | 69.5±0.1 | 58.1±0.1 | 57.4±0.0 | 59.7±0.1 |
| Random | 56.6±1.6 | 74.8±2.8 | 64.5±2.7 | 61.4±2.6 | 64.3±2.4 | 31.7±0.5 | 50.3±0.5 | 37.7±0.4 | 38.8±0.6 | 39.6±0.5 |
| Full Dataset | 92.5±0.0 | 95.2±0.1 | 94.1±0.1 | 89.4±0.3 | 92.8±0.1 | 78.7±0.0 | 83.0±0.0 | 81.7±0.1 | 76.5±0.0 | 80.0±0.0 |

Table 1: **Linear Probes with One Image-per-Class:** We compare our method (Distilled) to several real-image baselines on ImageNet-100 (**left**) and ImageNet-1k (**right**). Images are distilled (or selected) using the given model in each column. "Neighbors" are the real images with embeddings closest to those of our distilled images. "Centroids" are the real images with embedding closest to the mean of each class. "Random" is a random selection of real images. Our method outperforms each baseline across all models and both datasets.

| Distill Model | ImageNet-100 | | | | | ImageNet-1k | | | | |
|---|---|---|---|---|---|---|---|---|---|---|
| | CLIP | DINO-v2 | EVA-02 | MoCo-v3 | Average | CLIP | DINO-v2 | EVA-02 | MoCo-v3 | Average |
| CLIP | 84.9±0.1 | 80.8±0.4 | 83.8±0.2 | 61.6±0.2 | 77.8±0.2 | 63.0±0.0 | 56.4±0.1 | 59.7±0.1 | 39.5±0.0 | 54.7±0.1 |
| DINO-v2 | 77.0±0.1 | 91.5±0.1 | 86.8±0.1 | 78.8±0.1 | 83.5±0.1 | 54.1±0.0 | 75.0±0.1 | 65.4±0.1 | 60.0±0.0 | 63.7±0.1 |
| EVA-02 | 75.5±0.2 | 86.4±0.1 | 89.0±0.0 | 67.7±0.1 | 79.7±0.1 | 55.5±0.1 | 65.9±0.1 | 70.3±0.1 | 51.8±0.0 | 60.9±0.1 |
| MoCo-v3 | 65.6±0.1 | 86.6±0.1 | 82.3±0.2 | 83.4±0.1 | 79.5±0.1 | 41.4±0.0 | 66.9±0.1 | 57.2±0.1 | 63.2±0.0 | 57.2±0.1 |
| Full Dataset | 92.5±0.0 | 95.2±0.1 | 94.1±0.1 | 89.4±0.3 | 92.8±0.1 | 78.7±0.0 | 83.0±0.0 | 81.7±0.1 | 76.5±0.0 | 80.0±0.0 |

Table 2: **Cross-Model Performance of Distilled Datasets**: Here, we see ImageNet-100 (**left**) and ImageNet-1k (**right**) distilled using a given model and then evaluated across all models. We see that images distilled from DINO have the best average cross-model performance for both datasets. The distilled datasets generalize well, aside from an outlier pair of CLIP and MoCo. Columns are colored based on percentage of the "Full Dataset" benchmark.

## 4 Experiments

We evaluate our method on various datasets, including ImageNet-1k [12] and ImageNet-100 [42] for our primary results, Spawrious [26] and Waterbirds [38] for a study on adversarial datasets, Stanford Dogs [21] and CUB-200-2011 [45] for fine-grained visual classification, and ArtBench [24] to test the method's out-of-distribution capabilities. The majority of our experiments use four pre-trained self-supervised feature extractors as backbones: CLIP [34], DINO-v2 [31], EVA-02 [14], and MoCo-v3 [10]. In all our experiments, we distill the given dataset for 5000 iterations before training linear probes to convergence on the resulting synthetic images. All experiments are conducted at $224 \times 224$ resolution and use the "ViT-B" version of the given model. All distilled datasets by default use 10 sets of augmentations per batch except for ImageNet-1k, for which only 3 sets of augmentations are used due to compute constraints. For further implementation details, please see the Appendix.

**Evaluation and Baselines.** To measure our method's performance on a given feature extractor, we randomly initialize a linear classifier and optimize it to convergence using the distilled images before evaluating on the test set. The same procedure is used to evaluate real-image baselines. In prior works on self-supervised learning [31], the evaluation consists of a grid search across a number of hyper-parameters, including from which layer(s) the features used to train the linear probe should be taken from. For simplicity, we instead use only the features from the backbone's *last layer* and keep the training hyper-parameters consistent across the training of all linear probes.

We compare our method against three real-image baselines. For the **Neighbors** baseline, we choose the real image for each class whose *distill-model* embedding is closest to that of the corresponding synthetic image produced by our method. Similarly, for **Centroids**, we take the real training image with embedding closest to the *mean* embedding for each class. Lastly, for **Random**, we simply select a random image for each class and average the performance over 10 different seeds.

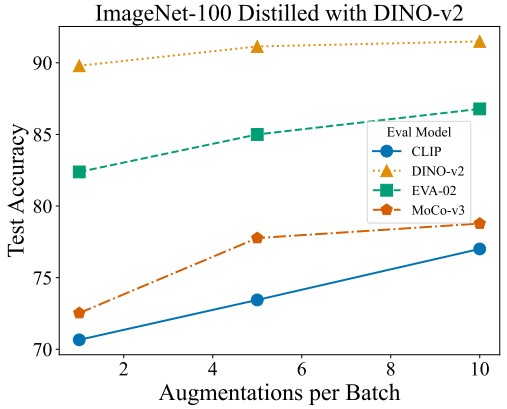

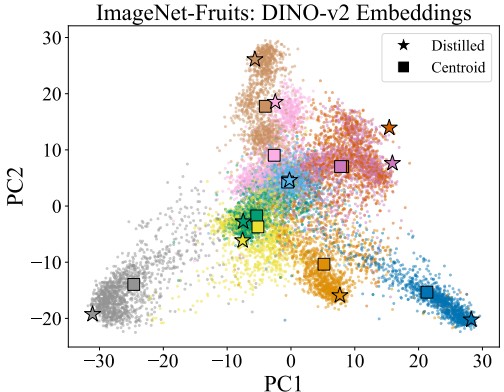

Figure 3: Performing more rounds of differentiable augmentation on the synthetic data during each distillation step improves both the single-model and cross-model performance of the distilled images.

Figure 4: We distill ImageNet-Fruits and observe the PCA of the training image embeddings. Each color represents a class. Note that the distilled images typically lie on the edge or outside of their class's cluster.

## 4.1 Linear Gradient Matching Out-Performs Real-Image Baselines

First, we evaluate our method on the original dataset distillation task: using a backbone model to distill a dataset and then training a model *of the same architecture* on the resulting synthetic images. Note that this setting differs from previous dataset distillation works in that we use a *pre-trained backbone* distillation and reuse that backbone during evaluation by training a new randomly-initialized linear classifier on top of it.

In Table 1, we see results for the single-model setting when distilling ImageNet-100 and ImageNet-1k. Across both datasets and all four models, **our method convincingly out-performs all real-image baselines**. In particular, our method enables a linear probe trained on top of DINO-v2 [31] to reach **75% test accuracy** on ImageNet-1k while only ever having seen a **single labeled image per class**. In comparison, training on the full dataset of 1.3 million real images reaches just 7 points higher at 83%. Similar results are seen for the other models and datasets as well.

In Figure 4 we visualize the *embeddings* of the distilled images relative to the real training data by plotting the 2D principal component analysis (PCA). For the sake of visual clarity, we use ImageNet-Fruits [6, 7], a toy dataset of 10 fruit classes from ImageNet-1k. The embeddings in this figure are from the same model used to distill the dataset (DINO-v2). We observe that the embeddings of the distilled images tend to lie far away from their respective class's centroid, often falling on the outside edge of the class's distribution. We hypothesize that this is due in-part to the distillation embedding *highly-discriminative* features within the synthetic images.

## 4.2 Distilled Images Generalize Across Models

Next, we investigate the cross-model generalization capabilities of our distilled datasets. That is, we first distill a dataset using one backbone model (e.g., CLIP) and then evaluate the synthetic images' performance on the *other* models (DINO-v2, EVA-02, MoCo-v3) that were not seen during distillation.

We visualize these results in Table 2. The diagonal elements are equivalent to the first row of Table 1 and represent single-model performance while the off-diagonals show cross-model capabilities. We see that the distilled datasets tend to generalize well to unseen models save for an outlier between CLIP and Moco, possibly due to poor model *alignment* (Section 4.6). We also observe that the highest-accuracy model on the full dataset (DINO) also has the distilled dataset with the best cross-model performance, suggesting that a model's quality affects the generalization capabilities of its distilled data.

## 4.3 Regularization and Augmentation Enable High Performance

In this section, we quantitatively (Figure 3) and qualitatively (Table 5) analyze the effects of the various regularization and augmentation techniques included in our distillation process. Table 3 shows the effects of ablating various components when evaluating using the same model used to distill (**top**)

| Train Set (1 Img/Cls) | ImageNet-100 | | | | |
|---|---|---|---|---|---|
| | CLIP | DINO-v2 | EVA-02 | MoCo-v3 | Average |
| **Same Eval** Full (Ours) | **84.9**±0.1 | **91.5**±0.1 | **89.0**±0.0 | **83.4**±0.1 | **87.2**±0.1 |
| -Decorrelate | 82.6±0.1 | 91.3±0.2 | 89.0±0.1 | 83.2±0.1 | 86.5±0.1 |
| -Pyramid | 83.5±0.2 | 91.0±0.3 | 87.9±0.1 | 80.5±0.1 | 85.7±0.1 |
| -Augment | 58.4±0.2 | 82.6±0.4 | 74.0±0.3 | 59.6±0.5 | 68.6±0.4 |
| **Avg Cross** Full (Ours) | **75.4**±0.3 | **80.9**±0.1 | 76.6±0.1 | 78.2±0.1 | **77.8**±0.2 |
| -Decorrelate | 69.4±0.4 | 79.5±0.1 | **76.8**±0.1 | **79.9**±0.2 | 76.4±0.2 |
| -Pyramid | 57.3±0.2 | 74.5±0.2 | 68.5±0.2 | 68.1±0.2 | 67.1±0.2 |
| -Augment | 35.3±0.4 | 31.6±0.2 | 34.3±0.5 | 32.2±0.5 | 33.3±0.4 |
| Full Dataset | 92.5±0.0 | 95.2±0.1 | 94.1±0.1 | 89.4±0.3 | 92.8±0.1 |

Table 3: **Evaluating Ablations:** While all three components provide improvements, the Augmentation has the most dramatic effect, especially in the cross-model setting. Likewise, the Pyramid optimization seems to matter more in the cross-model setting than the same-model setting by mitigating over-fitting to the model used during distillation.

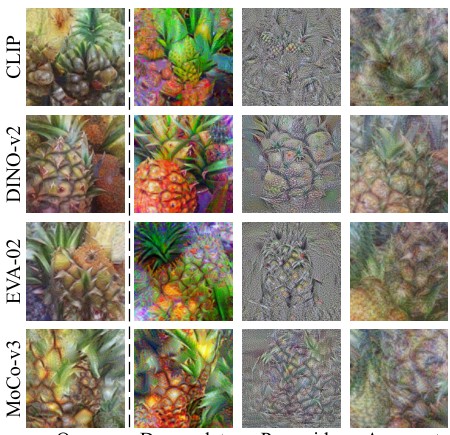

Figure 5: **Visualizing Ablations:** Removing various components of our pipeline causes visual degradation in the distilled images.

and averaging performances across the other three models (**bottom**). Figure 5 visualizes the distilled Pineapple class from ImageNet-100 under the same ablations.

**Color Decorrelation**  Inspired by prior work in feature visualization [30] and motivated by a desire to spare our synthetic images from any color-related biases imposed by the model used to distill them, we apply a color decorrelation technique to our distillation process. In Figure 5, we see that the images distilled without the color decorrelation (column 2) look less realistic than those using our full method (column 1); they are over saturated and contain high levels of incorrect colors (blue in this example). Quantitatively, however, this component has the lowest effect of the three analyzed in this section. While generally slightly helpful, it only offers a large improvement in the cross-model setting when distilling using CLIP.

**Pyramid Representation**  It has been shown in prior work [7] that pixel-based optimization for dataset distillation does not scale well to higher resolutions; the synthetic images tend to become riddled with high-frequency patterns that inhibit their usefulness as training data. As discussed in Section 3.2, we instead adopt a pyramid representation for the distillation process. In Figure 5, we see samples distilled from ImageNet-100 optimized using either the pyramid (column 1) or naïve pixel representation (column 3). We encourage the reader to zoom in and observe the high-frequency patterns and overall lack of coherence in the pixel-based images. In Table 3, we see that the images distilled without the pyramid representation do indeed make far worse training data in the cross-model setting, causing the model to overfit to the high-frequency patterns.

**Differentiable Augmentations**  Since its first proposal [53], the incorporation of differentiable augmentations during the distillation process has proven critical to the efficacy of the synthesized images as training data. As such, we also perform differentiable augmentations in this work, as described in Section 3.3. In Figure 5, we see samples from ImageNet-100 distilled both with (column 1) and without (column 4) augmentations. Visually, we observe that the images distilled *without* augmentations lack much meaningful structure and appear to simply be blobs of color with just a hint of geometry. In Table 3, we see that the lack of augmentations *severely* limits the synthetic images' usefulness as training data, even when evaluation on the same model used to distill.

As earlier discussed, we also apply *multiple rounds* of augmentations at each distillation step and concatenate the multiple augmented versions together to form the synthetic batch. Figure 3 illustrates how the number of *augmentations per batch* affects both the single-model and cross-model performance of the distilled data. Specifically, we see that when distilling ImageNet-100 with DINO-v2, raising the number of augmentations per batch from 1 to 5 and eventually 10 steadily increases the distilled images' effectiveness as training data both on the backbone model used to distill (DINO-v2) and the other unseen models (CLIP, EVA-02, MoCo-v3).

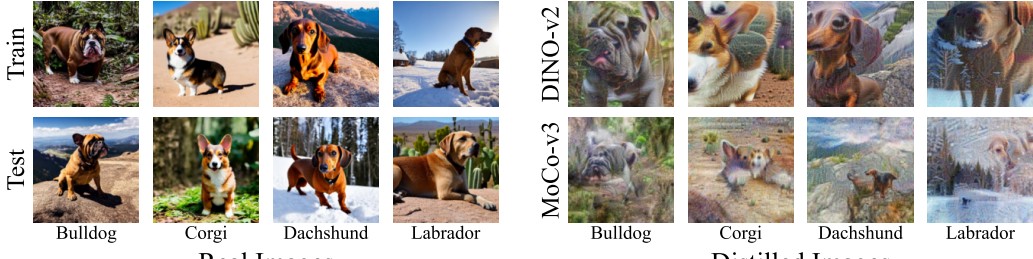

Figure 6: **Distilling Datasets with Spurious Correlations:** The 4-class "Spawrious" dataset contains spurious background correlations in the training set that are then subverted in the test set (**left**). A DINO linear probe trained on the full training set performs well, reaching 78% test accuracy, while a MoCo probe fails catastrophically, only reaching 36%. The distilled images (**right**) hint as to why: those distilled with DINO-v2 still contain mostly decipherable dog breeds while the MoCo-v3 counterparts focus almost entirely on the background environments. These images likely reflect the same biases held by the models used to distill them.

| Train Set (1 Img/Cls) | Spawrious | | | | | WaterBirds | | | | |
|---|---|---|---|---|---|---|---|---|---|---|
| | CLIP | DINO-v2 | EVA-02 | MoCo-v3 | Average | CLIP | DINO-v2 | EVA-02 | MoCo-v3 | Average |
| Distilled (Ours) | **43.1**±6.4 | **80.8**±3.1 | **36.5**±3.1 | **32.7**±2.3 | **48.3**±3.7 | **77.9**±0.2 | **82.1**±2.9 | **78.0**±0.8 | **77.8**±0.0 | **79.0**±1.0 |
| Neighbors | **41.7**±4.6 | **76.9**±1.2 | **40.7**±2.7 | **33.0**±3.4 | **48.1**±3.0 | **74.6**±6.5 | 67.3±5.8 | **74.3**±3.3 | 45.2±11.4 | 65.3±6.8 |
| Centroids | **44.8**±5.0 | **80.2**±2.7 | **38.1**±3.3 | **30.5**±1.6 | **48.4**±3.2 | 69.8±8.6 | 65.7±3.4 | 64.6±7.9 | 62.0±5.2 | 65.5±6.3 |
| Random | **46.2**±4.4 | 68.1±8.1 | **33.3**±5.3 | **31.8**±2.5 | **44.9**±5.1 | 71.9±4.5 | 57.1±8.6 | 59.8±13.3 | 67.4±5.7 | 64.0±8.0 |
| Full Dataset | 50.4±0.3 | 78.1±0.1 | 50.1±0.4 | 35.8±0.0 | 53.6±0.2 | 86.0±0.1 | 95.5±0.1 | 90.4±0.2 | 74.3±0.1 | 86.5±0.1 |

Table 4: **Performance on Datasets with Spurious Background Correlations:** The Spawrious (**left**) and Water-birds (**right**) training sets contain intentionally adversarial background correlations that are then subverted in the test sets. On Spawrious, our method no longer out-performs the real-image baselines as with the standard datasets (Table 1). This is perhaps due to the synthetic data adopting the models' biases and overfitting to the backgrounds on the training sets. We also see interesting interpretability results in the images themselves (Figure 6).

## 4.4 Distilling Adversarial Datasets Reveals Interpretable Model Weaknesses

The "Spawrious" dataset [26] consists of four classes of dog breeds and is designed to evaluate a model's ability to focus on the relevant content of an image rather than spurious correlations. In the training data, each breed is paired with its own unique environment: (Bulldog, Forest), (Corgi, Desert), (Dachshund, Mountain), and (Labrador, Snow). However, in the test set, the environments are intentionally different: (Bulldog, Mountain), (Corgi, Forest), (Dachshund, Snow), and (Labrador, Desert). We also experiment with a similarly constructed data, "Waterbirds" [38] consisting of just two classes (land birds and water birds) with similarly spurious background correlations.

Quantitative results on these datasets are shown in Table 4. Our method generally outperforms the real-image baselines, but to a significantly smaller extent than on the standard datasets (Table 1). We also observe that when training on the full Spawrious dataset, DINO-v2 [31] *far* out-performs the rest of the models (81% test accuracy) with MoCo-v3 [10] doing dramatically worse (37%), both to degrees far greater than seen when training on standard datasets (as in Table 1).

By *distilling* Spawrious with our method, we gain potential insights as to why we might see such dramatic results. As seen in Figure 6, the images distilled with DINO-v2 contain clear (albeit abstract) depictions of the correct dog breeds. On the other hand, those yielded by MoCo-v3 are almost entirely indecipherable, with the only clear portions being the spurious backgrounds from the training set. This interpretability result gives a clue towards explaining MoCo-v3's poor performance on this dataset; the model clearly seems to focus on the spurious background correlations present in the training set rather than the relevant subjects, causing catastrophic overfitting.

## 4.5 Distillation Excels in Fine-Grained Visual Classification

While computer vision methods are typically benchmarked against datasets containing broad categories of objects, such as ImageNet [12], much research also focuses on the task of Fine-Grained Visual Classification (FGVC) wherein datasets consist of many closely related classes that can be

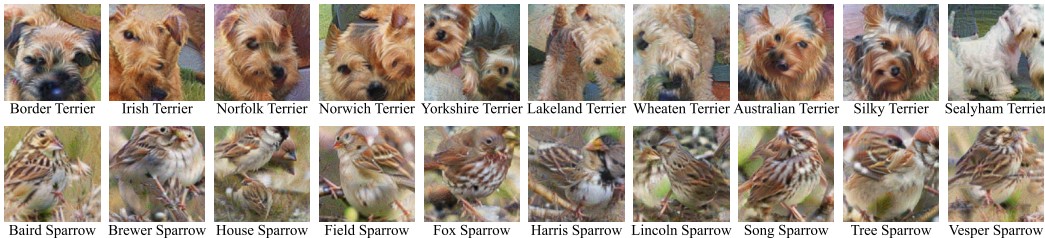

| Border Terrier | Irish Terrier | Norfolk Terrier | Norwich Terrier | Yorkshire Terrier | Lakeland Terrier | Wheaten Terrier | Australian Terrier | Silky Terrier | Sealyham Terrier |

| Baird Sparrow | Brewer Sparrow | House Sparrow | Field Sparrow | Fox Sparrow | Harris Sparrow | Lincoln Sparrow | Song Sparrow | Tree Sparrow | Vesper Sparrow |

Figure 7: **Distilling Fine-Grained Datasets:** Our distillation method captures the details necessary to teach a classifier to distinguish between highly similar classes. Pictured above are just 10 of the 120 classes distilled from the Stanford Dogs [21] dataset (**top**) and 10 of the 200 classes distilled from Caltech-UCSD Birds [45] using DINO-v2. For the full distilled datasets, please see the Appendix or our project page.

| Train Set (1 Img/Cls) | Stanford Dogs | | | | | CUB-2011 | | | | |
|---|---|---|---|---|---|---|---|---|---|---|
| | CLIP | DINO-v2 | EVA-02 | MoCo-v3 | Average | CLIP | DINO-v2 | EVA-02 | MoCo-v3 | Average |
| Distilled (Ours) | **52.1**±0.2 | **83.0**±0.1 | **74.8**±0.1 | **69.6**±0.2 | **69.9**±0.2 | **62.2**±0.2 | **86.0**±0.1 | **74.1**±0.2 | **42.5**±0.2 | **66.2**±0.2 |
| Neighbors | 33.4±0.1 | 71.3±0.2 | 58.5±0.2 | 56.3±0.1 | 54.9±0.2 | 39.4±0.1 | 76.9±0.0 | 52.6±0.3 | 28.1±0.0 | 49.2±0.1 |
| Centroids | 43.3±0.1 | 73.0±0.2 | 60.9±0.2 | 55.2±0.2 | 58.1±0.2 | 54.3±0.2 | 78.5±0.2 | 59.9±0.3 | 30.2±0.1 | 55.7±0.2 |
| Random | 23.3±1.5 | 51.9±1.8 | 38.3±1.8 | 36.6±1.4 | 37.5±1.6 | 37.5±1.6 | 64.4±1.5 | 44.3±1.5 | 19.1±0.5 | 41.3±1.3 |
| Full Dataset | 76.9±0.1 | 88.6±0.1 | 82.6±0.1 | 72.3±0.5 | 80.1±0.2 | 77.5±0.7 | 90.2±0.2 | 84.0±0.3 | 43.7±0.8 | 73.8±0.5 |

Table 5: **Performance on Fine-Grained Datasets:** Our distillation method captures the most discriminative aspects of each class, thereby enabling a down-stream classifier to correctly identify samples from datasets where all classes are closely related. In particular, the performance gap between our method and the real-image baselines is even higher on these fine-grained datasets (Stanford Dogs [21] and CUB-200-2011 [45]) than on the standard ImageNet benchmarks.

challenging for even human experts to distinguish. Such datasets require models to learn fine-grained features in order to identify the subtle differences between each class.

We apply our linear method to two common FGVC datasets: Stanford Dogs [21] and Caltech-UCSD Birds-200-2011 [45]. Samples of these datasets distilled using DINO-v2 can be seen in Figure 7. As seen in Table 5, in this more challenging fine-grained setting, our method outperforms the real-image baselines to a *significantly higher* degree than on the standard ImageNet benchmarks (Table 1). It seems as though the distillation process's ability to store didactic information in a single image per class is even more relevant in this setting since any one real image is less likely to contain the information necessary to distinguish its entire respective class from the other highly similar classes.

### 4.6 Cross-Model Performance Predicts Model Alignment

Recent research has suggested an "alignment" of various self-supervised models despite large differences in their training methods or even data modalities. The notion that all these models are indeed converging to a unified embedding space has been dubbed the "Platonic Representation Hypothesis" [20]. While the representations of today's models are not yet perfectly synchronized, it is still of interest to measure the degree to which they are aligned.

In the hypothesis's titular work, the authors introduce a method of measuring the alignment of two models called "mutual $k$ nearest neighbors." In short, this method finds each sample's $k$ nearest neighbors in the embedding spaces of each model and then computes the fraction of neighbors that are shared between the two.

When we distill images with a given model A and measure the 1 nearest neighbor performance (since there is just one image per class) on a different model B, we find that the normalized test accuracy is strongly correlated with the mutual $k$-nn alignment between models A and B, as seen in Table 6. This is an interesting interpretability result showing that the cross-model performance of a distilled dataset highly depends on the alignment between the two models. Given this dependency, our distillation method then acts as a method of *visualizing* how well these models align by backing the discrepancies in the embedding space out to image space. For example, given the two least aligned models (CLIP and MoCo), we can see in Figure 1 that even for different sets of classes, the two models clearly distill very different *styles* of images, alluding towards their misalignment.

| Distill Model | Normalized $k$-NN Accuracy | | | | | Source Model | Model Alignment | | | |
|---|---|---|---|---|---|---|---|---|---|---|
| | CLIP | DINO-v2 | EVA-02 | MoCo-v3 | | | CLIP | DINO-v2 | EVA-02 | MoCo-v3 |
| CLIP | | 69.5 | 72.3 | 46.6 | | CLIP | | 0.21 | 0.26 | 0.18 |
| DINO-v2 | 77.2 | | 95.5 | 91.7 | | DINO-v2 | 0.21 | | 0.39 | 0.31 |
| EVA-02 | 66.1 | 86.1 | | 70.2 | | EVA-02 | 0.26 | 0.39 | | 0.30 |
| MoCo-v3 | 57.2 | 87.4 | 76.8 | | | MoCo-v3 | 0.18 | 0.31 | 0.30 | |

Table 6: **Distilled datasets predict model alignment.** We distill ImageNet-1k and find the synthetic data's *cross-model* performance (**left**) by evaluating on a model other than the one used during distillation. We find that this cross-model performance correlates well with the *alignment* between models' embedding spaces (**right**). Note the similarity of the per-row trends between the two tables. Rows are colored from highest to lowest.

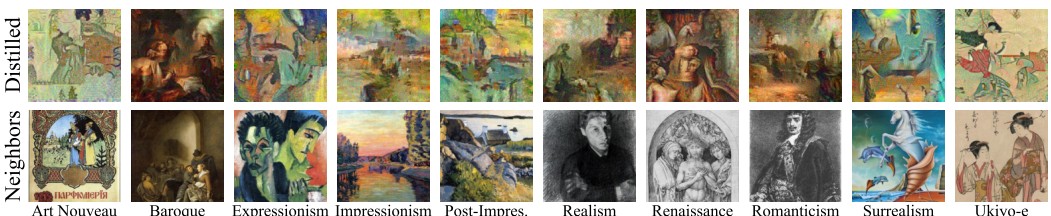

Art Nouveau  Baroque  Expressionism  Impressionism  Post-Impres.  Realism  Renaissance  Romanticism  Surrealism  Ukiyo-e

Figure 8: **Distilling Out-of-Distribution Datasets:** Despite only ever being trained on the real-world images in ImageNet-1k [12], our method can still use DINO-v1 [5] to distill out-of-distribution datasets such as ArtBench [24] that have no overlapping content. We see that the distillation is not simply "copying" samples from the dataset by comparing to the nearest real neighbor in the model's embedding space and observing the stark differences. This interpretability insight granted by our method highlights DINO's remarkable ability to generalize beyond its training distribution.

## 4.7 Self-Supervised Backbones can Distill Out-of-Distribution Datasets

Note that while we have thus far been discussing DINO-v2, this section in particular will reference DINO-v1 since this version's training set is open-sourced. Since DINO-v1 [5] was only trained on ImageNet [12], a dataset of nearly all "real-world" images, we can easily test its ability to distill an **out-of-distribution** dataset such as ArtBench [24] that consists of 10 classes of art styles. In Figure 8, we visualize the distilled images (**top**) as well as the nearest neighbor in DINO-v1's embedding space. Surprisingly, despite having previously only ever seen "real-world" images from ImageNet, DINO-v1 is still able to effectively distill ArtBench into a single image per class. Furthermore, we confirm that the distilling is not simply "copying" real images from the training set by observing the stark differences between the synthetic images and their nearest neighbors. This is especially apparent in the cases of Realism, Renaissance, and Romanticism where the nearest neighbors are black and white while the distilled images are colorful.

## 5 Conclusion

In this work, we introduce a new task in the area of *Dataset Distillation*: learning a tiny set of synthetic images designed to train **linear probes** on top of **pre-trained self-supervised vision models**. Our proposed solution, *Linear Gradient Matching*, optimizes a meta gradient objective to ensure that our synthetic images induce training updates similar to those obtained from the real data. Quantitatively, our method outperforms all baselines, enabling, for example, a DINO-v2 linear probe to reach 75% test accuracy on ImageNet-1k while only having ever seen **one labeled image** per class.

We also showcase the importance of secondary aspects of our Linear Gradient Matching method, such as the pyramid representation and differentiable augmentations, and highlight their importance in learning highly efficacious distilled datasets. Furthermore, we evaluate our method on challenging fine-grained datasets and show that it out-performs the real-image baselines by an even larger margin than on the standard ImageNet benchmarks. Lastly, our method yields several interesting interpretability results, such as giving insights into how these pre-trained models "see," predicting how well different models align, or elucidating a model's ability to generalize beyond its training distribution.

We hope our work brings to light the interesting set of problems posed by this new task and inspires others to continue in this area. Code and distilled datasets can be found on our project page.

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

# Appendix

To begin the appendix, we first speak on our method's limitations, broader impact, and compute budget. In the following sections, we include more information that was omitted from the main paper due to space constraints. In Section A, we provide a more detailed description of our method's implementation details. In Section C, we include visualizations of complete distilled datasets.

**Limitations**  Our method is largely limited by memory and data loading. For example, we were limited in our ImageNet-1k experiments to only using 3 rounds of augmentations per batch due to a data loading bottleneck. We hope to eventually develop a method that does not require the loading of thousands of real images per step. Additionally, due to the bi-level optimization, we are limited to using Pytorch's [2, 33] `nn.DistributedParallel`, rather than `nn.DistributedDataParallel`, which causes significant slowdowns due to our very large batch sizes. Using a different automatic differentiation framework, such as Jax [4], might alleviate this issue, but this would also require porting all our self-supervised backbones to the new framework.

**Broader Impact**  While our method quantitatively out-performs the real image baselines, the broader impacts likely lie in the introduction of this new task as well as the resulting images acting as an interpretability tool. As shown in the paper, the distilled images offer insights into things such as model alignment and how different networks see the world in varying ways.

Furthermore, the ultimate goal of dataset distillation is the reduction of training time, which equates to less energy spent. A linear probe trained on our distilled datasets can be trained to convergence in just a few minutes while training on the full dataset can take up to an entire day.

**Compute Budget**  We used a variety of GPUs for this work depending on what was available on the shared cluster. Specifically, we used a combination of H200, A100, L40s, Ada6000, and 4090 GPUs. Distilling ImageNet-100 with the default settings takes about 3 hours using 1 H200 GPU, and ImageNet-1k takes about 12 hours using 4 H200 GPUs.

# A  Implementation Details

In this section, we provide a more detailed explanation of our methodological and experimental implementation details.

## A.1  Distillation

We implement our method in Pytorch [2, 33], which has a unique license to which we comply.

We optimize our pyramid representations using Adam [22] with a learning rate of 0.002.

We distill for 5000 iterations and add a new level to the pyramids every 200 iterations until we reach the maximum resolution (256).

Each level of the pyramid is initialized with $\mathcal{N}(0, 1)$ normalized by the current number of levels in the pyramid. When a new level is added, the existing levels are appropriately re-normalized.

At each distillation iteration, we first sample a new linear classifier consisting of a weight matrix $W$ and bias vector $\mathbf{b}$ using the default initialization for Pytorch's `nn.Linear`. For simplicity, we omit the bias in the main paper.

Then, we reconstruct the synthetic images from their pyramid representations using the following equation (copied from the main paper for convenience):

$$X = \frac{1}{2} + \frac{1}{2}\tanh\left(\sum_{r \in \rho} \text{resize}_{256}(P_r)\right) \quad (4)$$

The synthetic images are then augmented several (10 by default) *different* (not sequential) times and concatenated together to form the full synthetic batch. We obtain our synthetic loss ($\ell_{\text{syn}}$) by passing this batch through the feature extractor $\phi$ and linear classifier ($W, \mathbf{b}$) and then compute the gradient of the synthetic loss with respect to the linear classifier (and bias). These gradients are then vectorized.

For our real batch, we sample a number of real images equal to the length of the *full* synthetic batch (i.e., after the different augmentations and concatenation) and augment them once (since the batch size already matches that of the full synthetic batch). We then get the real loss and gradients via the same procedure. The meta loss is then calculated as the cosine distance between the real and synthetic vectorized gradients.

## A.2   Augmentation

For our differentiable augmentations, we use a custom re-implementation of `torchvision` transforms that run on GPU and support different random transforms for each element in a batch. We initially used the Kornia [36] implementations, but found that they cause CPU bottlenecks with very large batch sizes. We use `RandomHorizontalFlip` with default parameters, `RandomResizedCrop` with `size=(224, 224)`, and `RandomGaussianNoise` with `std=0.2`.

## A.3   Evaluation

**Linear Probes.**   To evaluate a given distilled dataset (or selected coreset for the real baselines) on a given target model, we first randomly initialize a new linear classifier. We then train the linear classifier for 1000 epochs with a batch size of 100. We use an Adam optimizer with a learning rate of $0.001/256$ (taken from the DINO-v1 [5] evaluation protocol) along with a cosine decay learning rate schedule. We stop training early if the test accuracy has not improved over the last 50 epochs. For the training set, we perform the same set of augmentations as during distillation (horizontal flip, random resized crop, and Gaussian noise). The output of the random resized crop is of size $224\times224$. For the test set, we resize the shortest side to 256 and then do a center crop of $224\times224$.

**Nearest Neighbors.**   For our nearest neighbor evaluation, we first find the train set embeddings by resizing the shortest side of each test image to 256, taking a $224\times224$ center crop, and passing through the feature extractor. We then do the same for the test set and find the training embedding closest to each test embedding (by cosine distance) and report the class of said neighbor. We only do one nearest neighbor since we distill down to one image per class.

## A.4   Datasets

For ImageNet [12] and ImageNet [42], we build on the Torchvision [43] implementation as part of the Pytorch [2, 33] ecosystem. ImageNet has a unique non-commercial license, to which we have agreed.

Spawrious [26] is taken from the paper's official GitHub repository and uses a CC0-1.0 license.

Waterbirds [38] is taken from the WILDS [23, 39] benchmark. Both the Waterbirds and WILDS repositories use an MIT license.

Stanford Dogs [21] is taken from the project website. It presumably uses the same license as ImageNet [12].

Caltech-UCSD-200-2011 is taken from the project website. No license is given.

ArtBench [24] is taken from the paper's GitHub repository. The repository itself uses an MIT license. The data is sourced from WikiArt [49], Ukiyo-e Search [35], and The Surrealism Website [1], all of which have Fair Use licenses.

Flowers-102 [29] is taken from Torchvision. No license is given.

Food-101 [3] is taken from Torchvision. No license is given.

## A.5   Models

For CLIP [34], we use the official GitHub repository which has an MIT license.

For DINO-v1 [5], we use the official GitHub repository which has an Apache-2.0 license.

For DINO-v2 [31], we use the official GitHub repository which has an Apache-2.0 license.

For EVA-02 [13], we use the Pytorch Image Models [48] implementation hosted on Hugging Face [19] which has an MIT license.

| Train Set | Flowers-102 | | | | | Food-101 | | | | |
|---|---|---|---|---|---|---|---|---|---|---|
| (1 Img/Cls) | CLIP | DINO-v2 | EVA-02 | MoCo-v3 | Average | CLIP | DINO-v2 | EVA-02 | MoCo-v3 | Average |
| Distilled (Ours) | **79.5**±1.2 | **99.6**±0.0 | **97.8**±0.1 | **56.4**±3.1 | **83.3**±1.1 | **78.7**±1.6 | **83.7**±0.3 | **83.9**±0.2 | **31.6**±3.2 | **69.5**±1.3 |
| Neighbors | 69.4±0.8 | 99.4±0.2 | 90.8±0.2 | 48.6±1.8 | 77.0±0.7 | 58.8±1.3 | 73.8±0.4 | 71.5±0.3 | 25.2±1.3 | 57.3±0.8 |
| Centroids | 77.5±0.7 | 99.4±0.1 | 95.6±0.3 | 48.2±2.6 | 80.2±0.9 | 74.1±0.6 | 75.6±0.1 | 76.4±0.4 | 22.7±2.0 | 62.2±0.8 |
| Random | 67.9±1.1 | 98.8±0.4 | 91.0±0.8 | 36.8±2.2 | 73.6±1.1 | 48.5±1.0 | 52.8±1.9 | 49.9±0.6 | 13.5±1.7 | 41.2±1.3 |
| Full Dataset | 93.4±0.1 | 99.7±0.0 | 98.9±0.0 | 82.2±0.1 | 93.5±0.1 | 91.9±0.0 | 92.7±0.1 | 92.0±0.0 | 78.3±0.0 | 88.7±0.0 |

Table B.1: **Additional Datasets:** Distilling Flowers-102 [29] and Food-101 [3] show similar results; our distilled dataset consistently out-perform the real-image baselines.

For MoCo-v3 [10], we use the official GitHub repository which has a Creative Commons Attribution-NonCommercial 4.0 International Public License.

# B   Additional Results

In Table B.1, we include results for Flowers-102 [29] and Food-102 [3]. We see results following the same trends as the rest of the paper; the distilled datasets out-perform the real-image baselines across the board.

# C   Visualizing Full Datasets

In Figures C.1, C.2, C.3, and C.4 we present the entirety of ImageNet-100 [42] distilled with CLIP [34], DINO-v2 [31], EVA-02 [14], and MoCo-v3 [10] respectively. In Figures C.5–C.14, we show ImageNet-1k [12] distilled with DINO-v2 and omit the other models for the sake of brevity (and ink).

We also include Stanford Dogs [21] (Figure C.15), CUB-200-2011 [45] (Figures C.16 and C.17), Flowers-102 [29] (Figure C.18), and Food-101 [3] (Figure C.19) distilled with DINO-v2 and once again omit the other models. All distilled datasets, including those not inculded here, can be viewed on our project page.

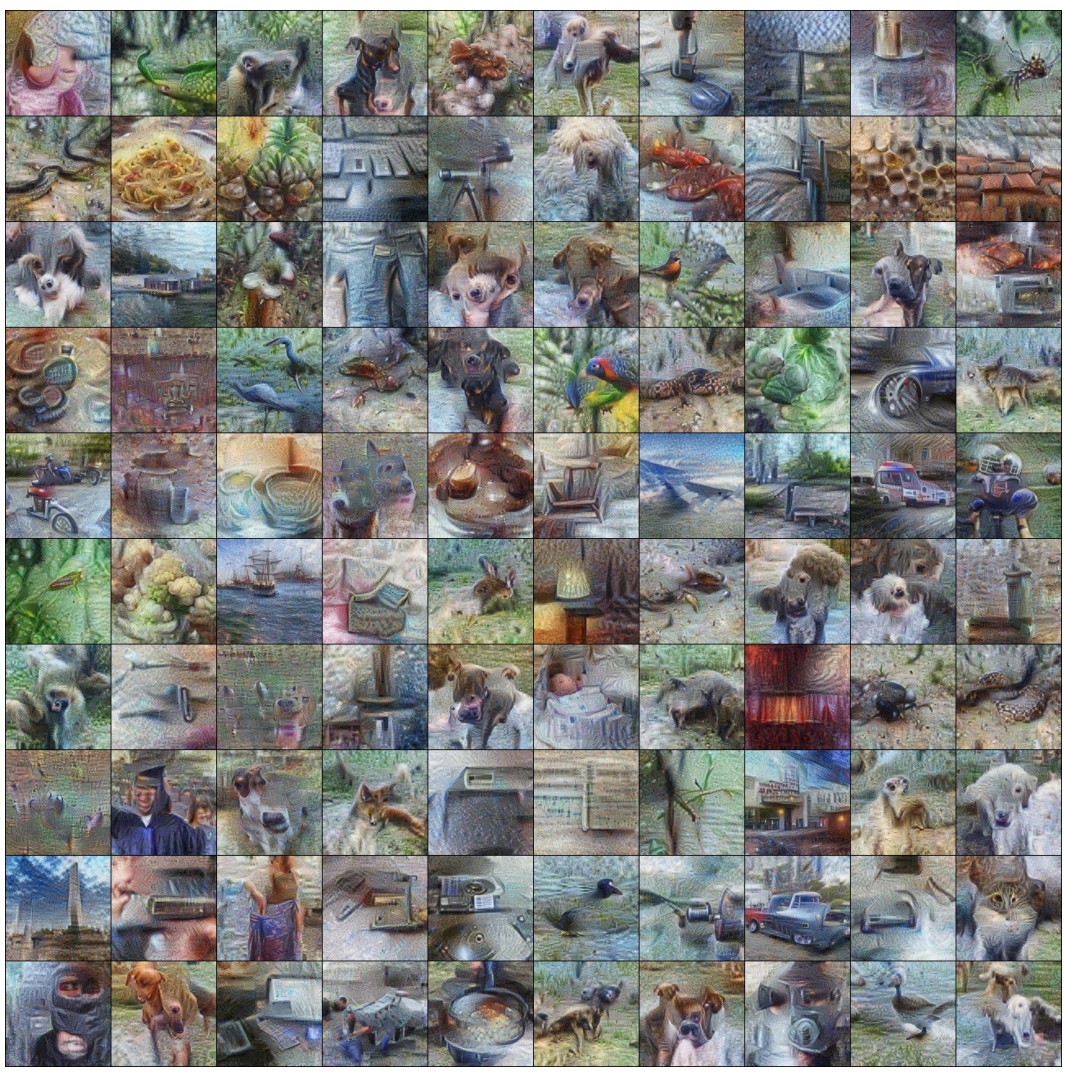

Figure C.1: ImageNet-100 [42] distilled using CLIP [34]

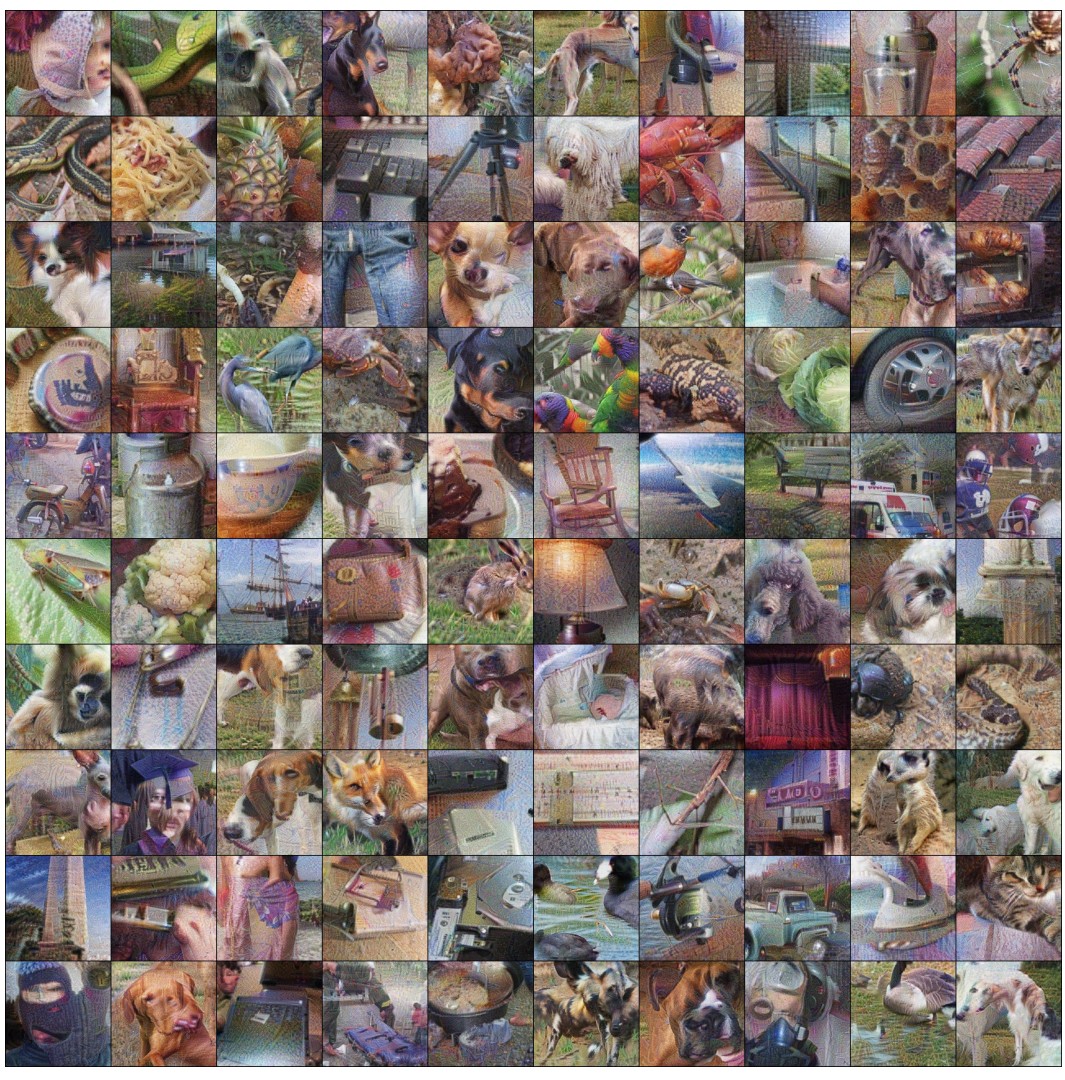

Figure C.2: ImageNet-100 [42] distilled using DINO-v2 [31]

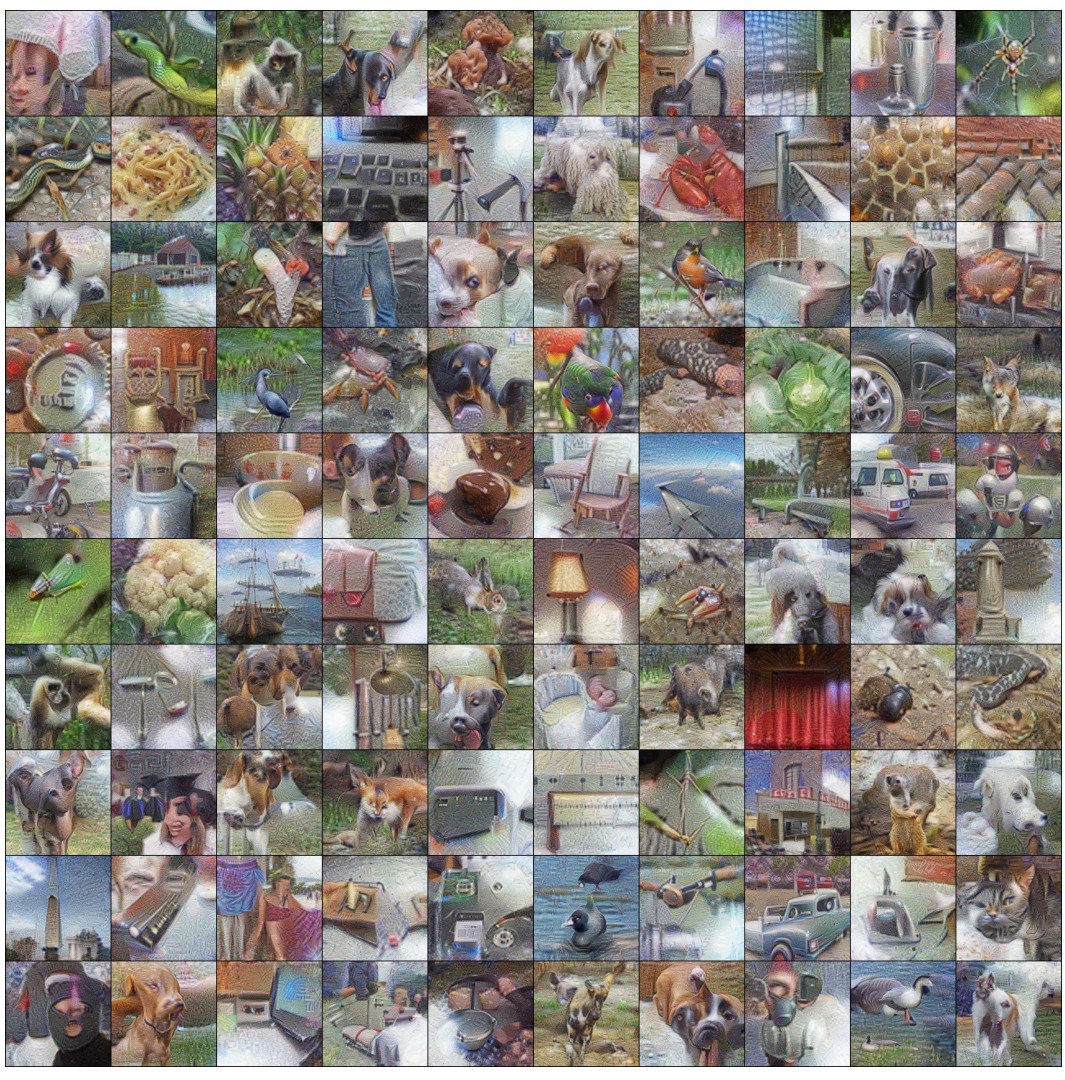

Figure C.3: ImageNet-100 [42] distilled using EVA-02 [13]

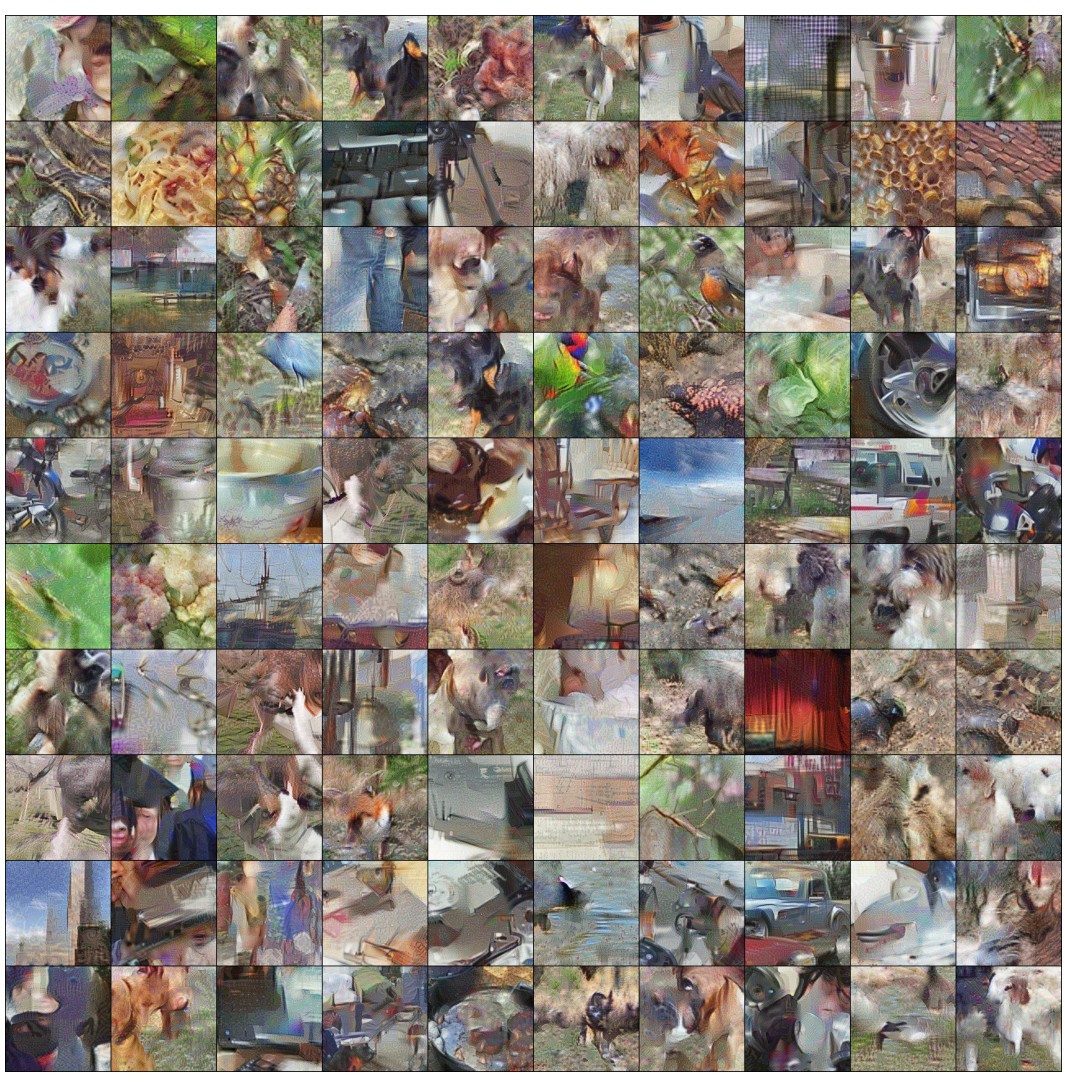

Figure C.4: ImageNet-100 [42] distilled using MoCo-v3 [10]

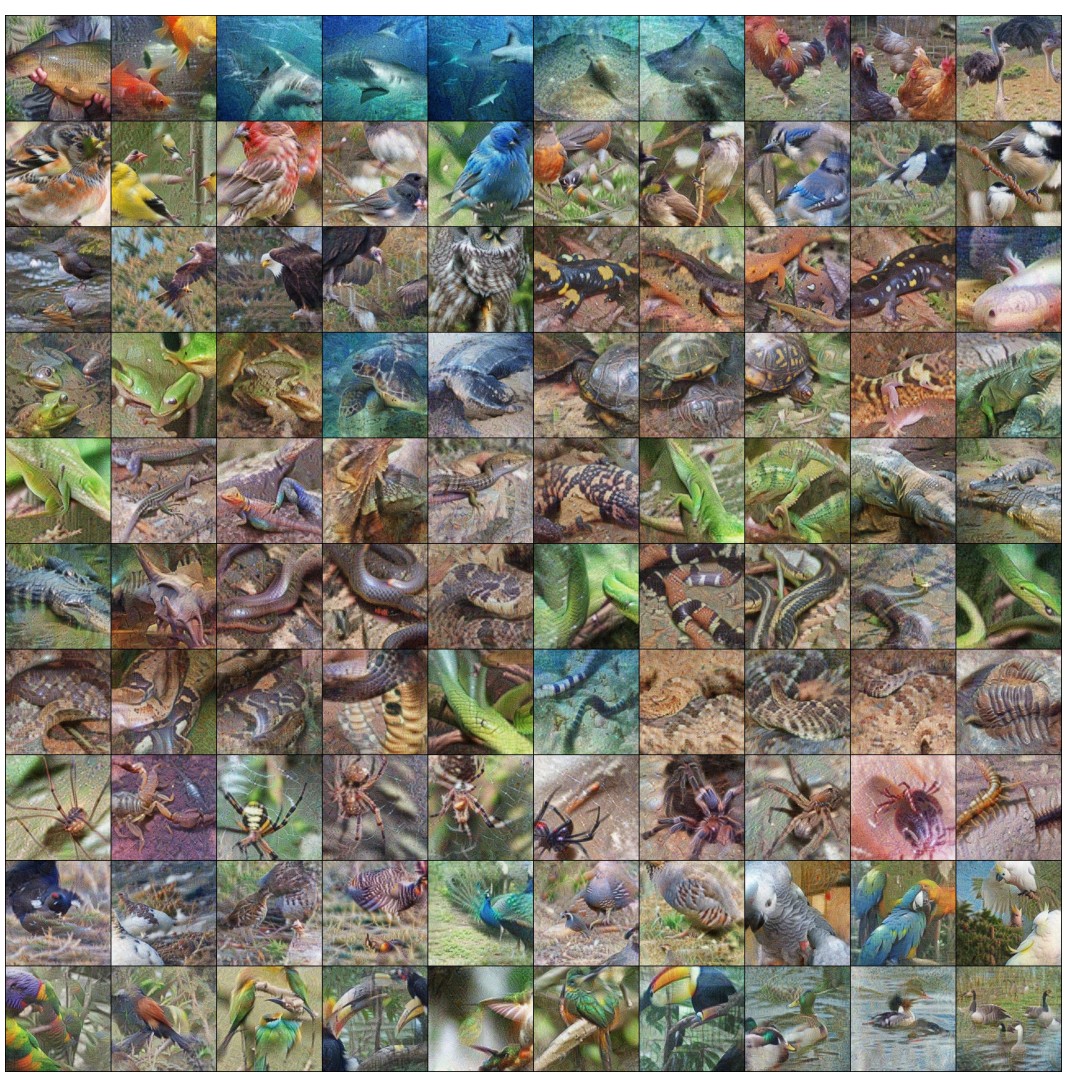

Figure C.5: ImageNet-1k [12] distilled with DINO-v2 [31] classes [0-99]

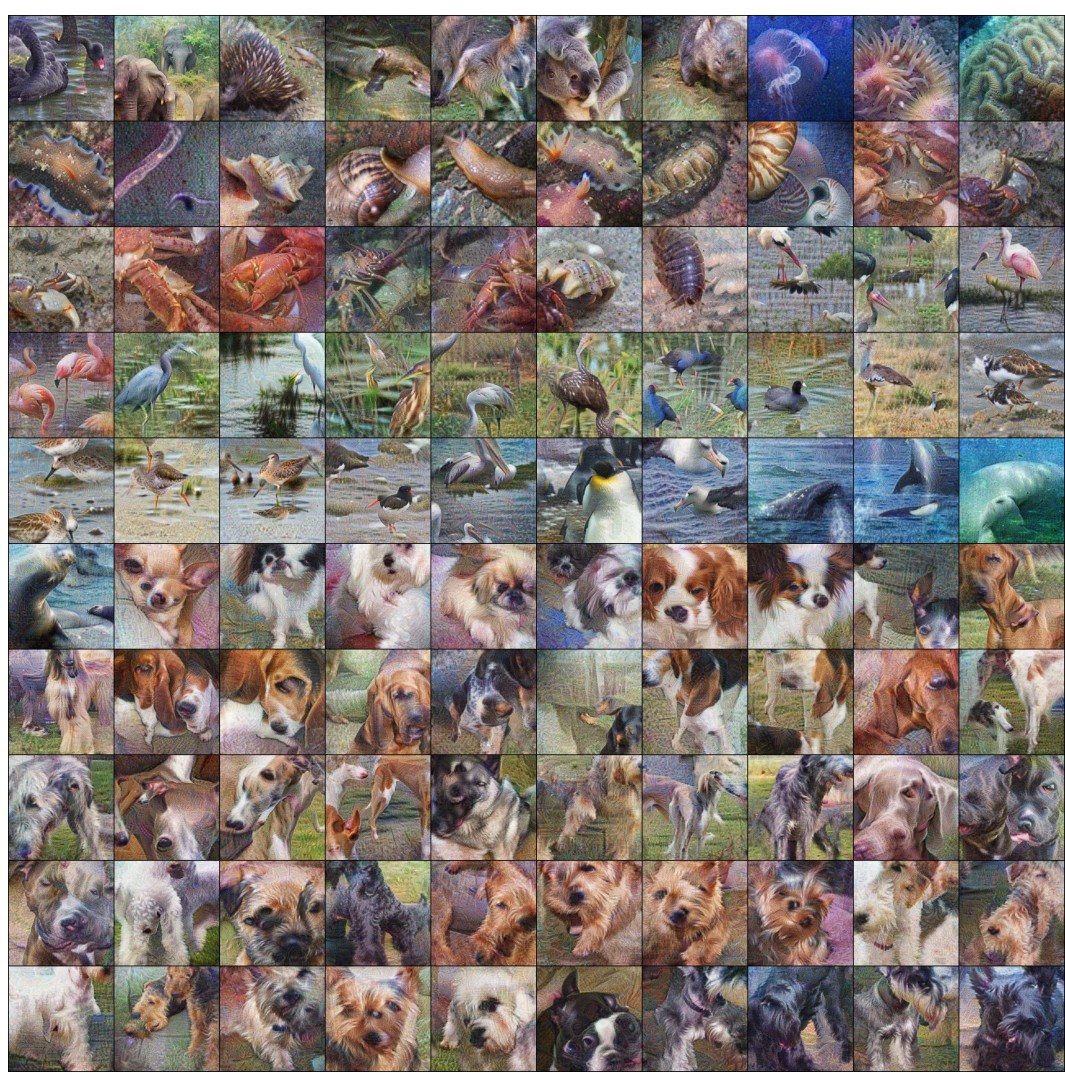

Figure C.6: ImageNet-1k [12] distilled with DINO-v2 [31] classes [100-199]

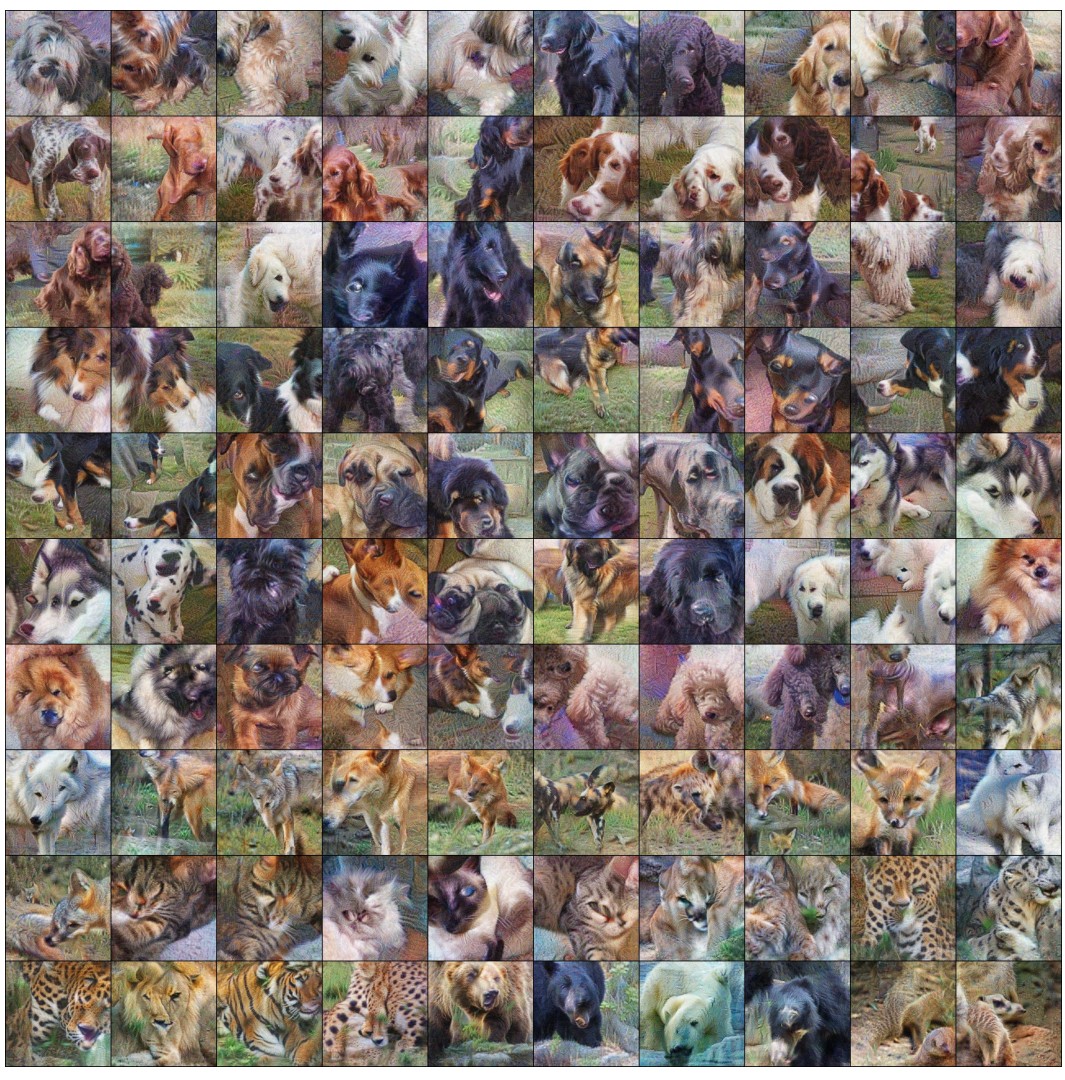

Figure C.7: ImageNet-1k [12] distilled with DINO-v2 [31] classes [200-299]

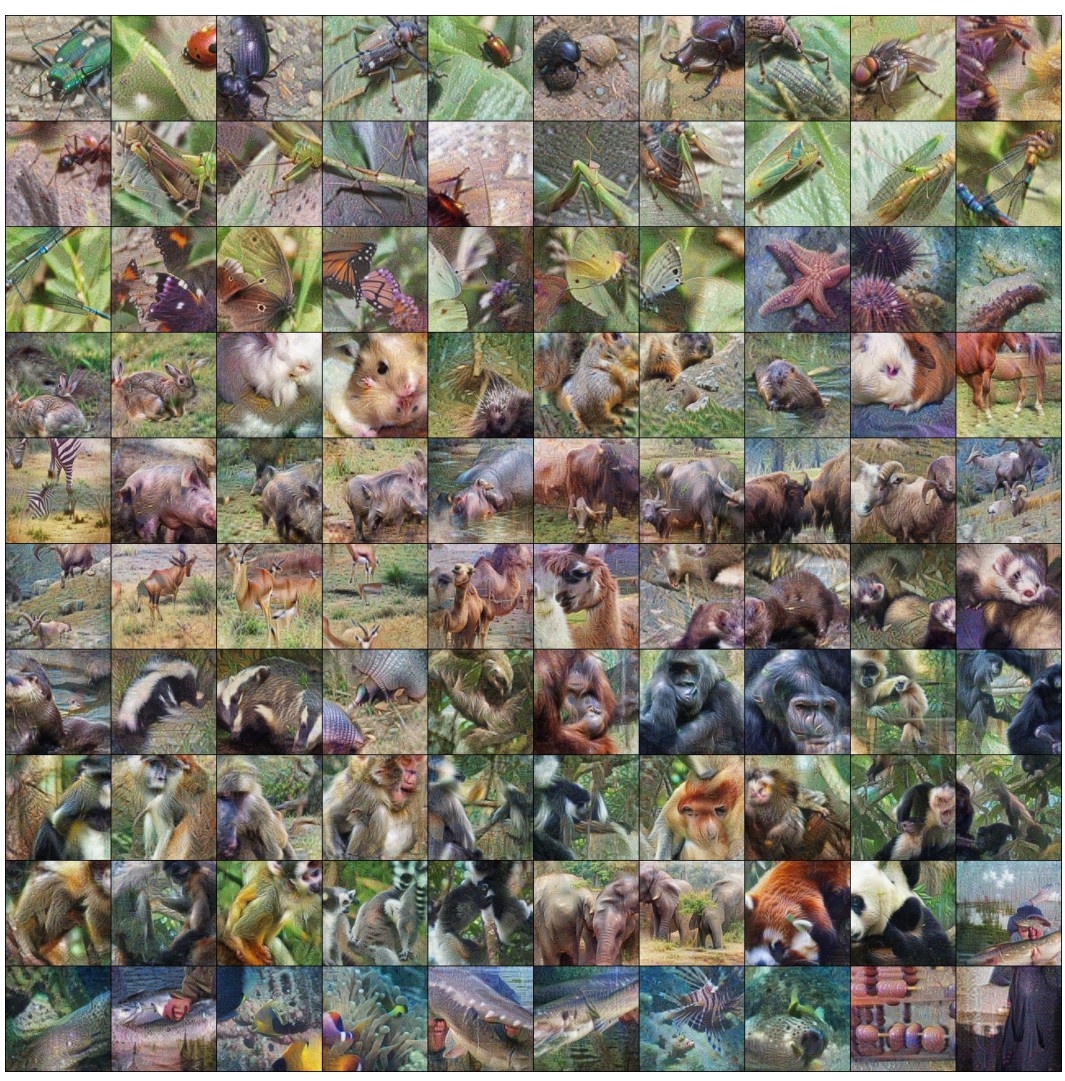

Figure C.8: ImageNet-1k [12] distilled with DINO-v2 [31] classes [300-399]

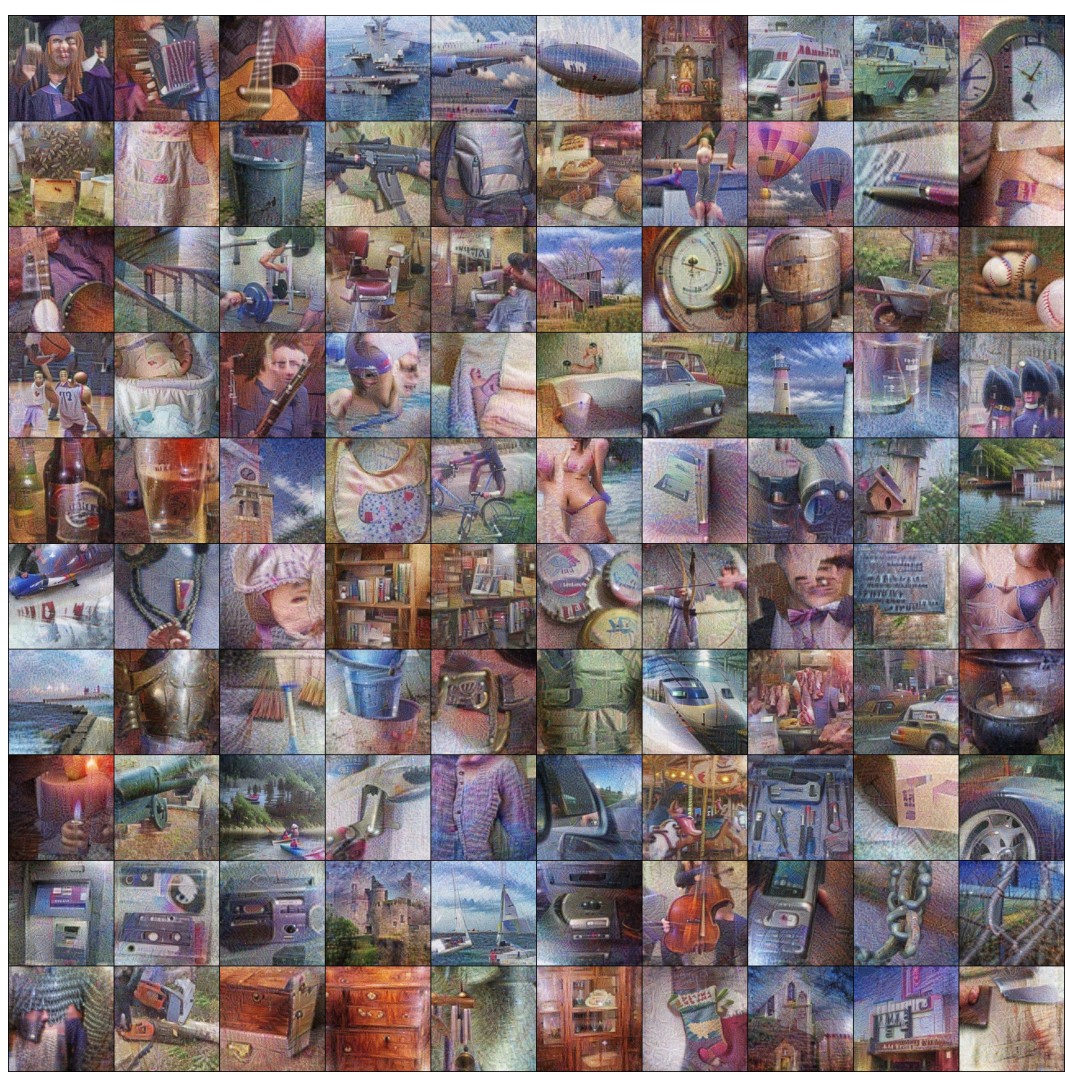

Figure C.9: ImageNet-1k [12] distilled with DINO-v2 [31] classes [400-499]

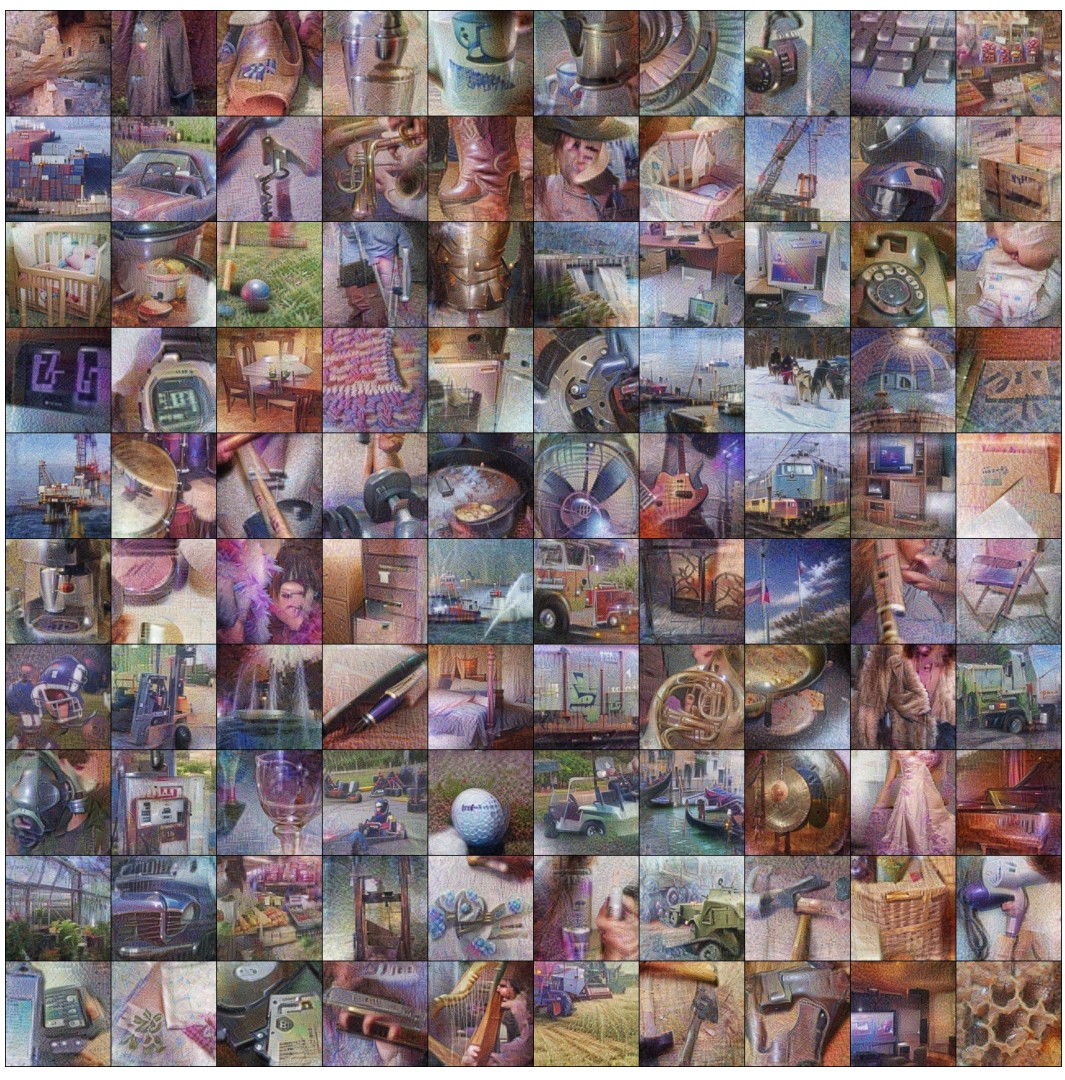

Figure C.10: ImageNet-1k [12] distilled with DINO-v2 [31] classes [500-599]

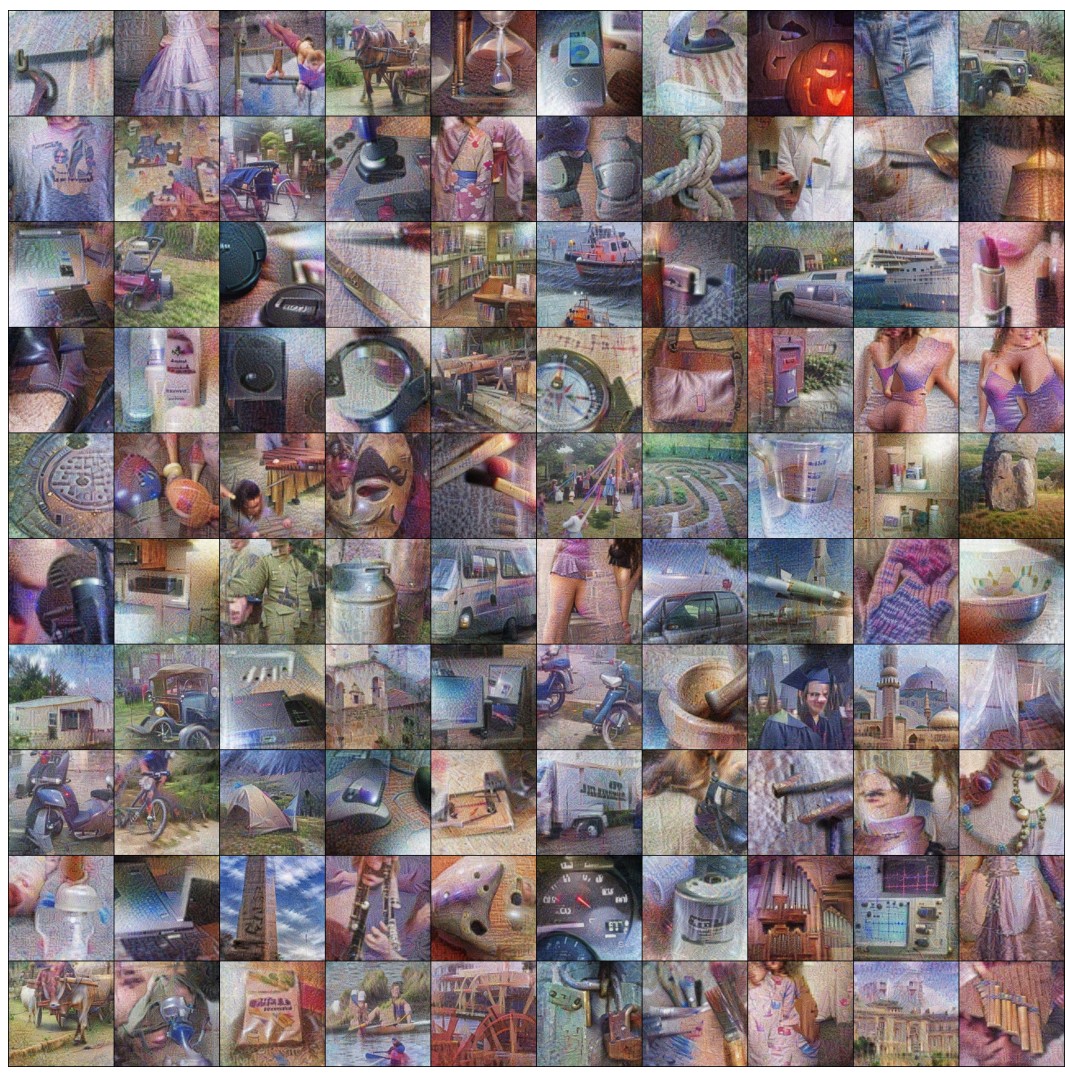

Figure C.11: ImageNet-1k [12] distilled with DINO-v2 [31] classes [600-699]

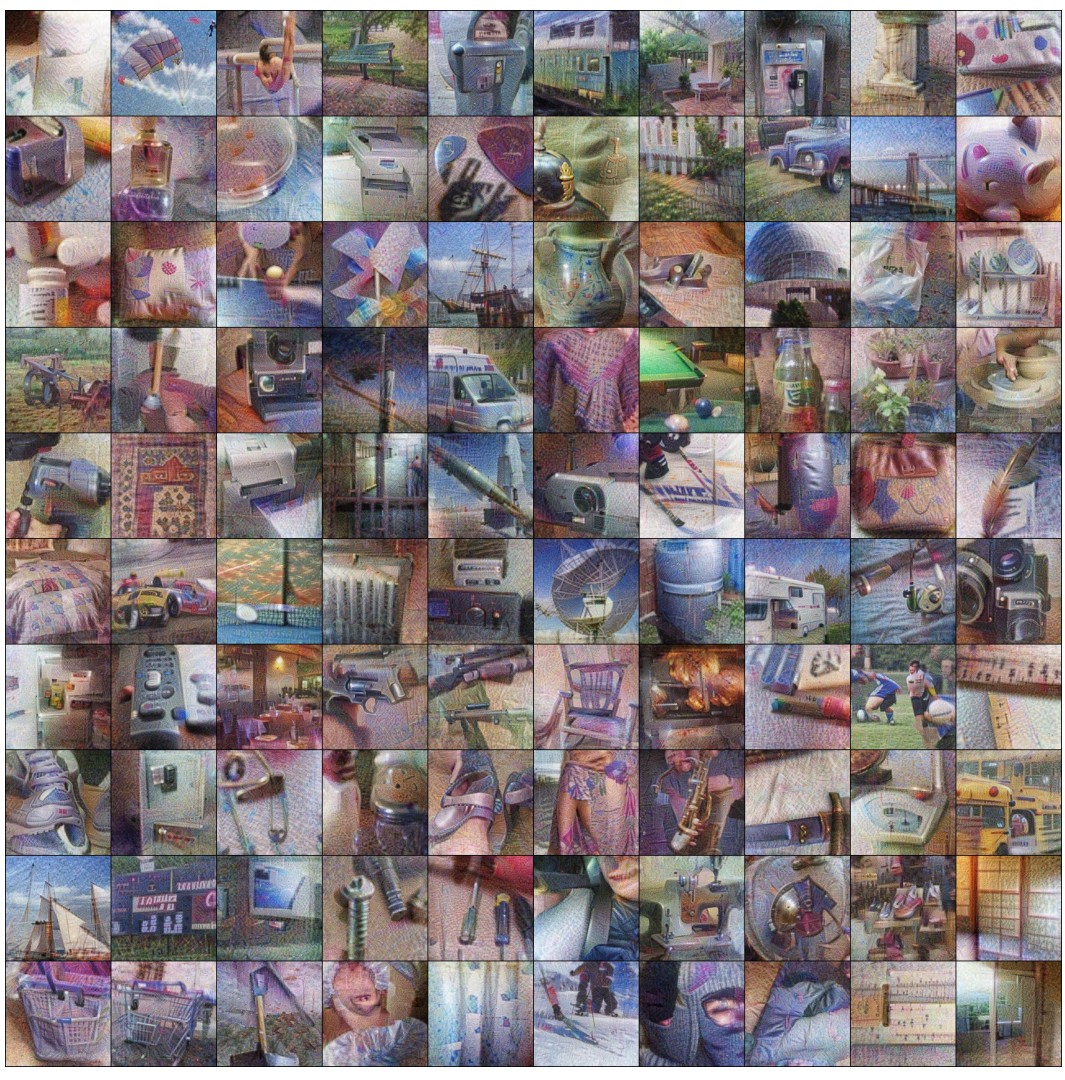

Figure C.12: ImageNet-1k [12] distilled with DINO-v2 [31] classes [700-799]

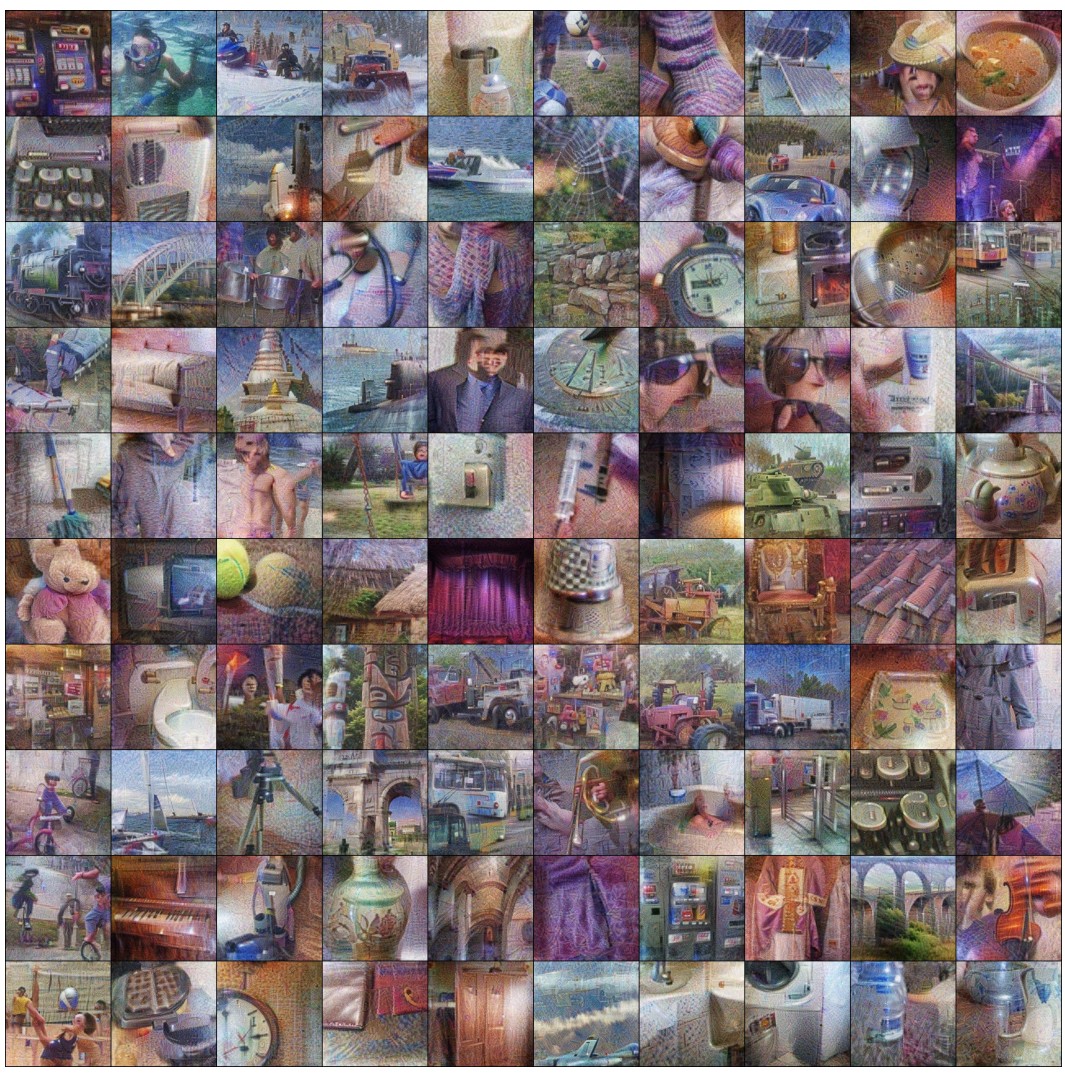

Figure C.13: ImageNet-1k [12] distilled with DINO-v2 [31] classes [800-899]

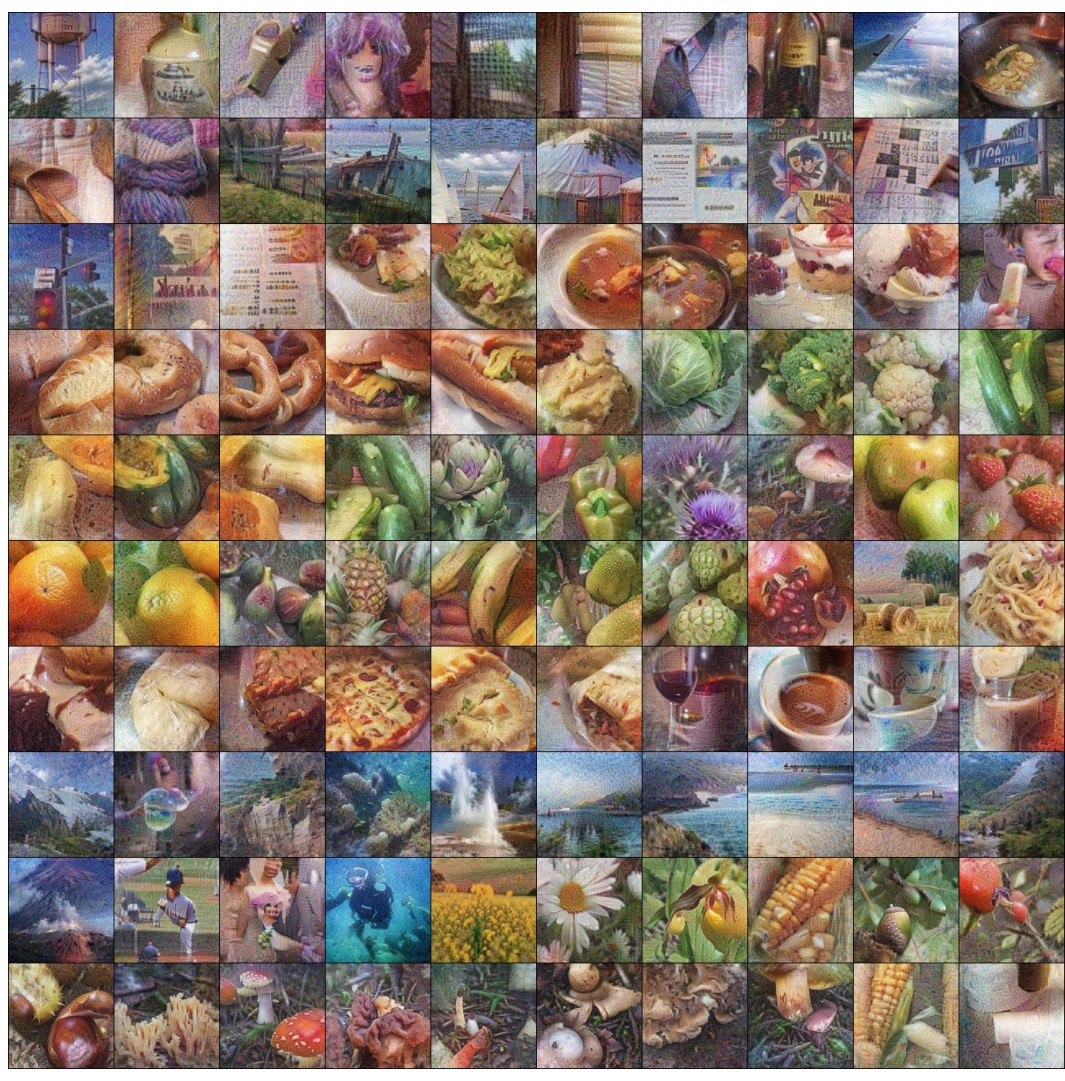

Figure C.14: ImageNet-1k [12] distilled with DINO-v2 [31] classes [900-999]

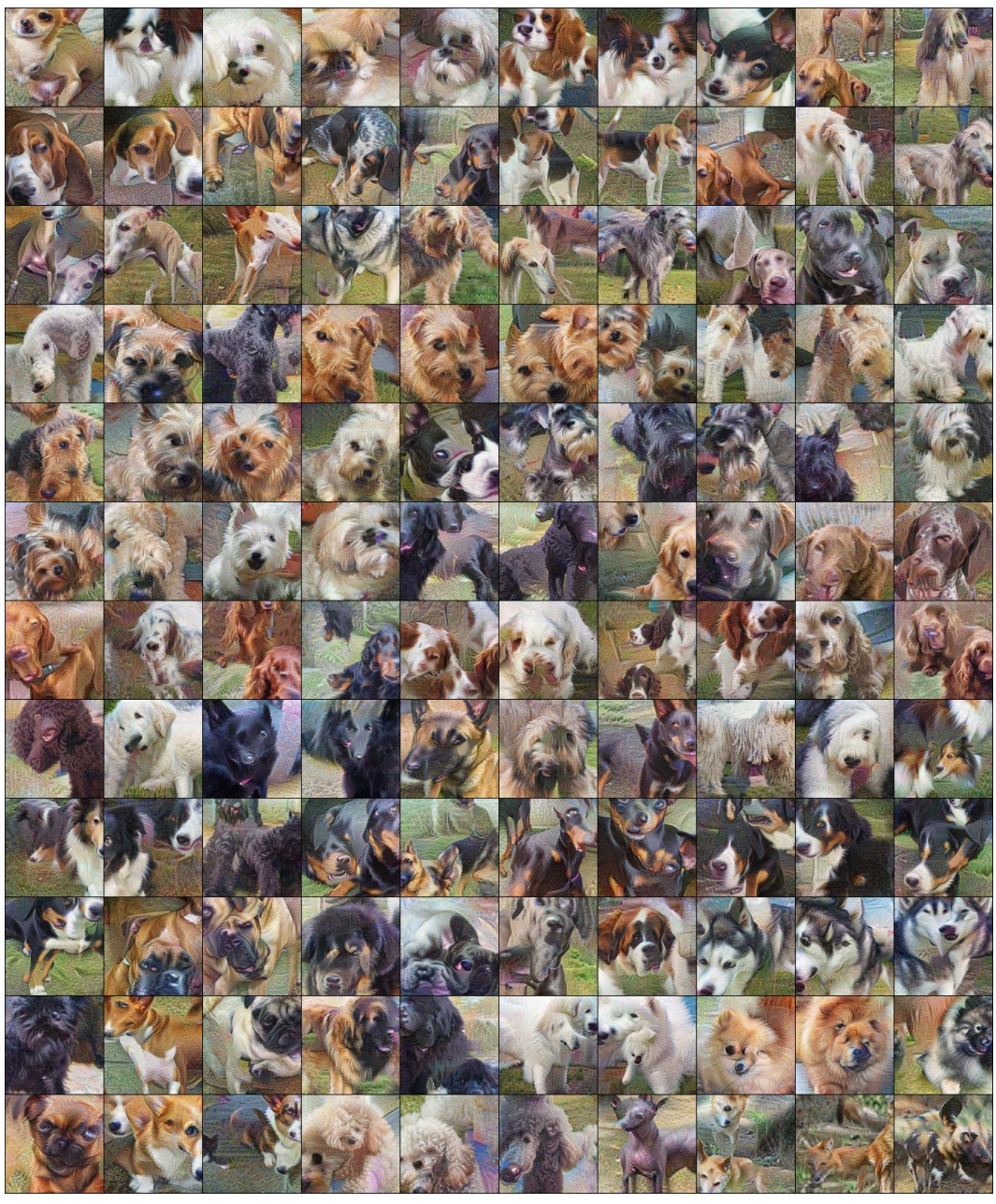

Figure C.15: Stanford Dogs [21] distilled using DINO-v2 [31]

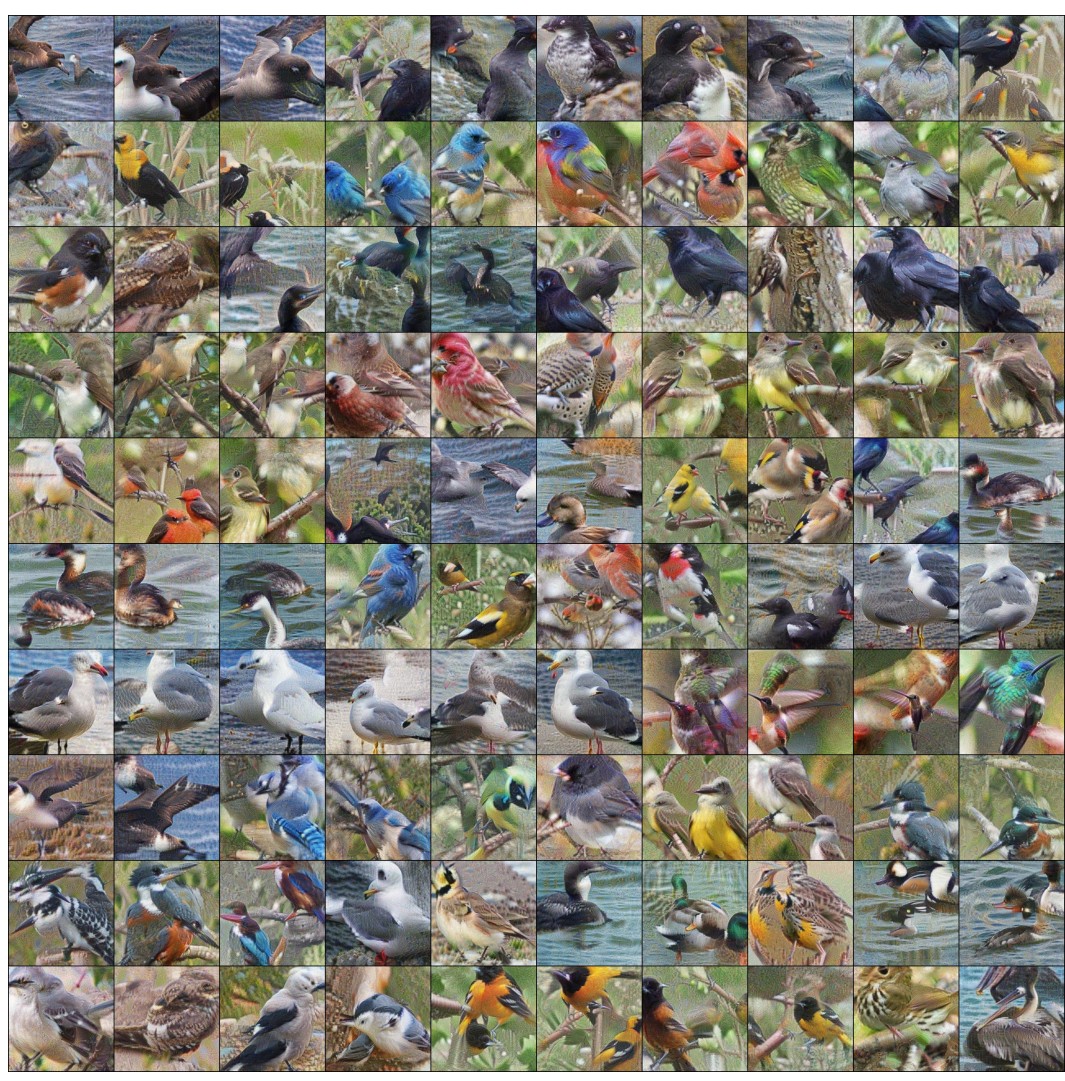

Figure C.16: CUB-200-2011 [45] distilled using DINO-v2 [31] classes [0-99]

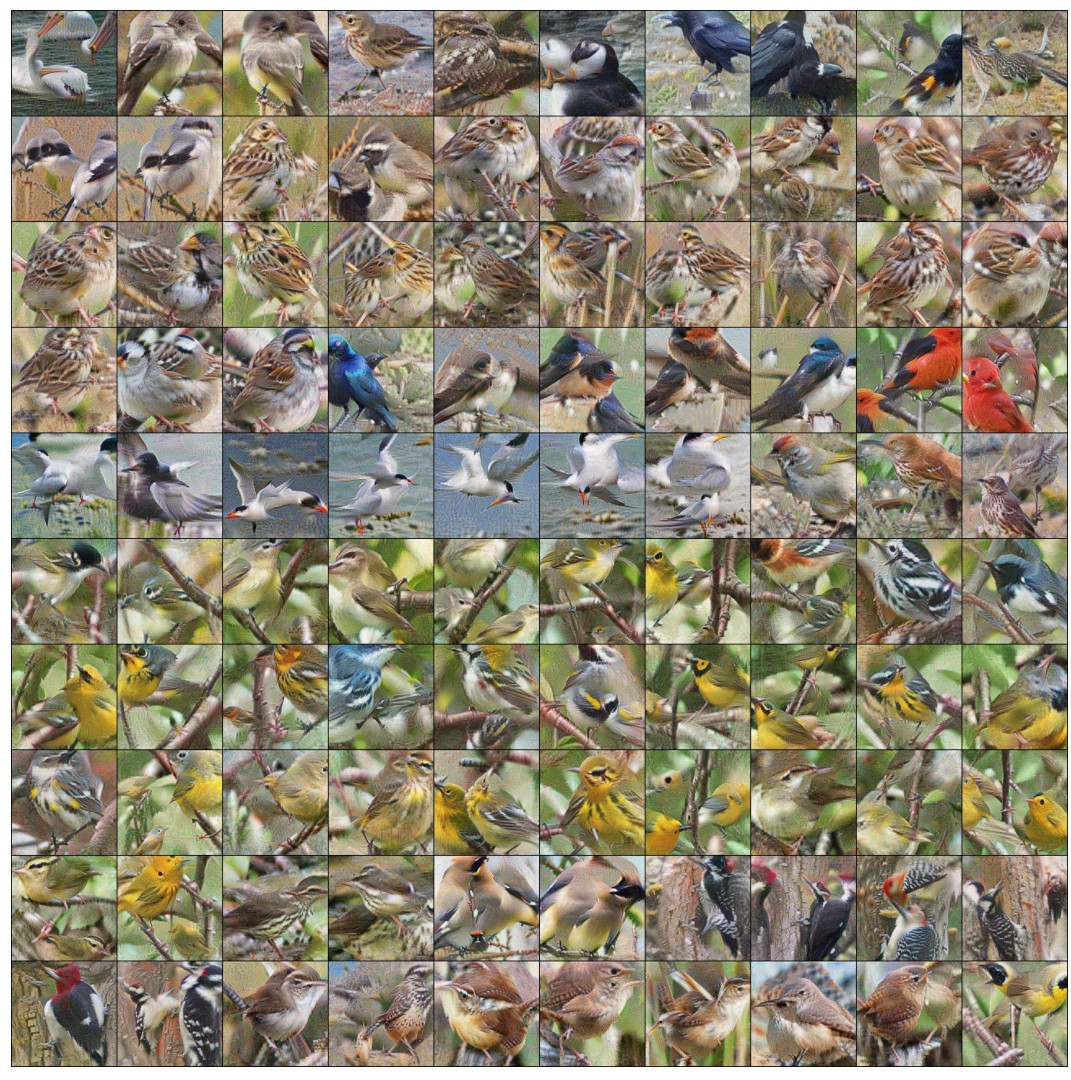

Figure C.17: CUB-200-2011 [45] distilled using DINO-v2 [31] classes [100-200]

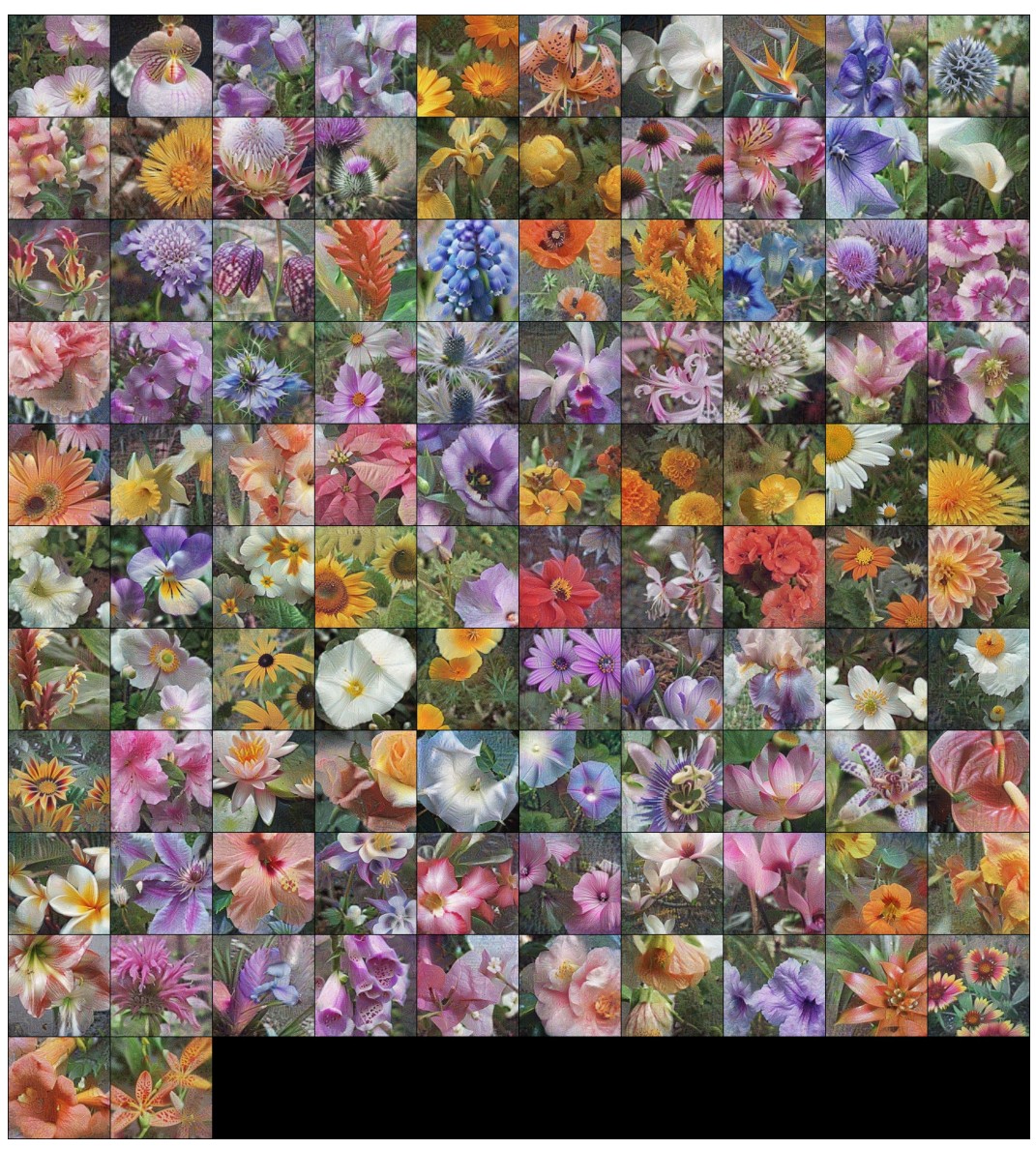

Figure C.18: Flowers-102 [29] distilled using DINO-v2 [31]

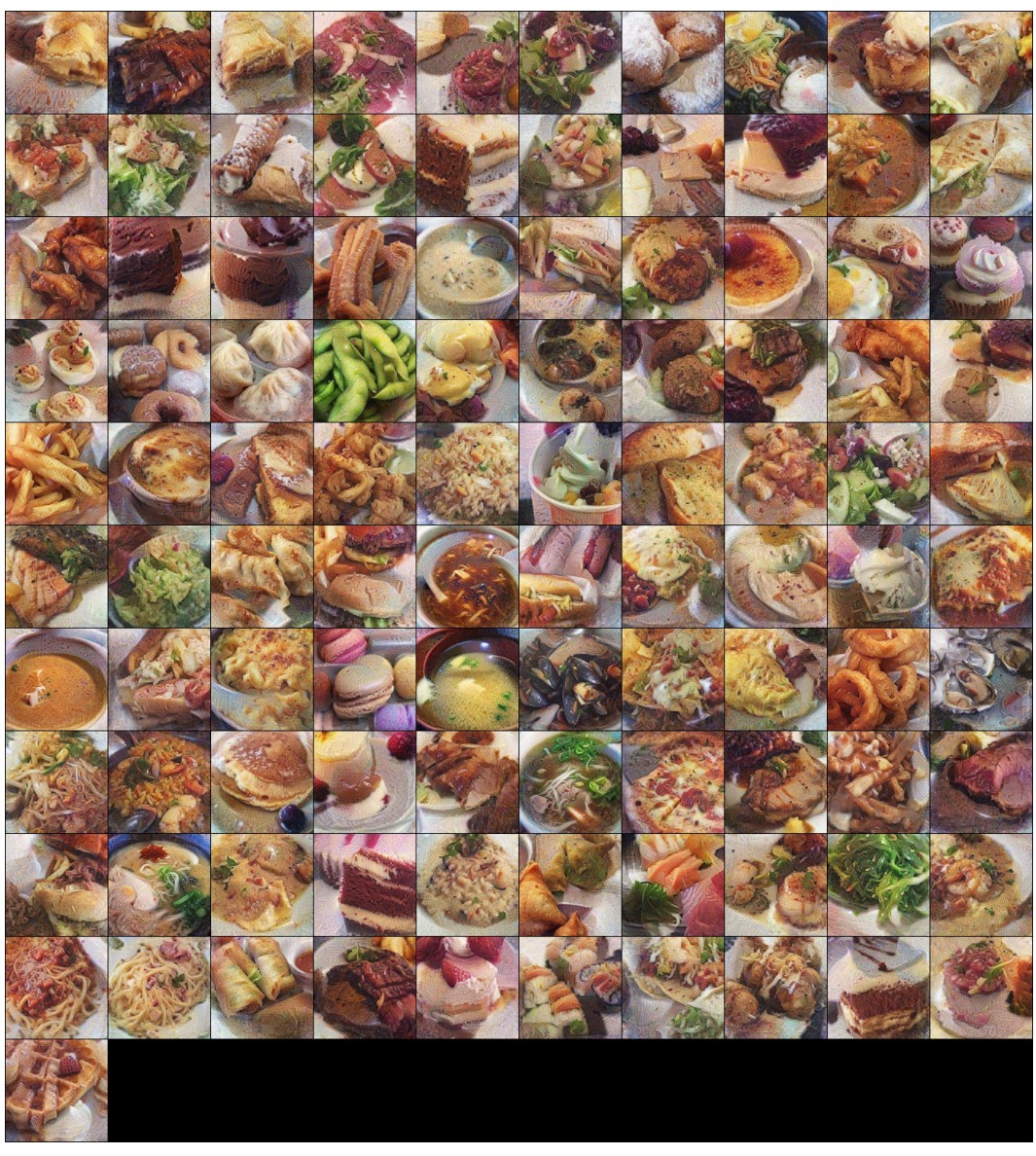

Figure C.19: Food-101 [3] distilled using DINO-v2 [31]

