# OpenReview forum: "Dataset Distillation for Pre-Trained Self-Supervised Vision Models"
_NeurIPS.cc/2025/Conference — NeurIPS 2025 poster_

### Official Review · Reviewer_gLBa · 2025-06-03

**Clarity:** 3
**Significance:** 3
**Originality:** 3
**Rating:** 4
**Confidence:** 5

**Summary:**

Summary

This paper addresses the performance of dataset distillation when the models to be trained are continued on pretrained models, e.g., linear probe, instead of from scratch. They propose to match: the gradient of loss wrt linear classifier when trained on [real, synthetic] data. The real/syn images are stored in different resolutions to avoid overfitting.

Experiments

Across 4 SSL methods, including CLIP and DINO, on ImageNet, linear probe on distilled dataset is better than other dd methods, and comparable to full data.

**Questions:**

How is your approach different from directly applying GM to linear probing? What about other trajectory matching methods?

**Ethical Concerns:**

["NO or VERY MINOR ethics concerns only"]

**Final Justification:**

Thank you for the clarification in the discussion.

**Limitations:**

The methodology seems to be the same as gradient matching. Are there any challenges when applying GM, or any other gradient based DD approaches for linear probing? Otherwise the technical contribution might be limited.

**Paper Formatting Concerns:**

looks good to me

**Quality:**

4

**Strengths And Weaknesses:**

Strengths: The paper is clearly written and experiment results are well presented. Finetuning might be a more prevalent task than training ilnear classification, imo.

Weaknesses: The technical contribution is limited as it directly applies gradient matching https://arxiv.org/abs/2006.05929 to another scenario, linear probing.

---

> ### Author Rebuttal · Authors · 2025-07-31
>
> Thank you for taking the time to review our paper.
>
> It seems your main concern is a potential lack of novelty, citing the "Dataset Condensation" paper by Bo Zhao et al.
> As discussed in our paper, this work was indeed a major inspiration for our Linear Gradient Matching method.
> However, it should be noted that this method and those that came after it (trajectory matching, etc.) generally fail to scale up to practical models and realistic datasets.
>
> It is true that our method, like that of Zhao et al., primarily functions by matching the gradients induced by real and synthetic gradients.
> However, our method makes several core improvements that allow it to scale, allowing us to distill practically useful datasets at just a single image per class.
> Furthermore, our method also serves as an interpretability tool, allowing researchers to better understand why different vision models behave in certain ways.
>
> Scaling up gradient matching in and of itself would be a novel contribution, but we additionally offer a number of new improvements unique to our work.
>
> To list some specific points of novelty:
> ### 1. Task-Specific Gradients.
> The original method of Zhao et al. optimizes each class of distilled images independently.
> This is particularly problematic in the 1 image-per-class setting (where we focus) since they do not even update the teacher network between iterations.
> By optimizing each class independently, the learning process misses out on the crucial information necessary to distinguish each class from *each other*.
> We jointly optimize across all classes such that the distilled images learn to induce the optimal N-way decision boundary between the classes.
>
> ### 2. Multiple Simultaneous Augmentations.
> To the best of our knowledge no existing dataset distillation method has employed our method of multiple augmentations.
> By creating a "mega batch" by combining several independently augmented versions of the distilled data, the synthetic images receive significantly better gradients.
> Rather than the optimizer forcing a single augmentation of the distilled image to be the "ideal" exemplar for that class,
> multiple simultaneous, disjoint augmentations allow for the joint set of augmented versions to serve as the informative class representatives together .
>
> ### 3. Pyramid Representation.
> Previous works have explored alternative representations for distilled data, but they typically either add a tremendous
> amount of overhead or reduce quality in the name of compression.
> To our knowledge, ours is the first work to represent the distilled data as a Laplacian pyramid of different resolutions.
> This unique parameterization implicitly regularizes the distilled data for only a minimal amount of extra computation.
> Without the pyramid representation, we would not be able to distill ImageNet or other high-resolution datasets to this level of quality.
>
> ### 4. Model Interpretability.
> By distilling datasets in the space of a pre-trained model, we gain valuable insights into that model's behavior.
> Despite coming from largely the same transformer architecture, the images distilled from our 4 different feature
> extractors differ quite significantly in appearance.
> This offers us insights as to how these models actually "see" and how they behave differently from each other.
> We further explore this in the section on spurious backgrounds and identify possible reasons as to why, for instance, MoCo gets fooled far more often than DINO.
>
> ### 5. The Task Itself.
> Dataset distillation (especially at 1 image per class) has largely been seen as a toy problem with limited practical uses.
> While achieving decent results on CIFAR and toy networks, existing methods failed to scale up to realistic settings.
> Given that most models today start from a pre-trained backbone, extending dataset distillation to this regime was a natural step.
> In doing so, and through the development of our new method, we can now train a linear classifier to nearly 75% test
> accuracy on ImageNet-1k in a matter of minutes using just a single image per class, as opposed to the 12+ hours and over
> 1 million images it would take to train using the full, real dataset.
>
> ### To answer some specific points you mentioned:
>
> > Finetuning might be a more prevalent task than training ilnear classification, imo.
>
> One of dataset distillation's points of failure is overfitting to the synthetic data.
> In the low image count regime (i.e., 1 image per class) networks easily overfit, leading to a large performance gap versus training on the full dataset.
>
> This overfitting is already a known issue with the toy ConvNets traditionally used in dataset distillation works,
> but training on a large pre-trained vision transformer heavily amplifies the problem.
>
> We present a table of results obtained by fine-tuning the models on the distilled data from ImageNet-100.
> Indeed, we observe overfitting to an extreme degree, with CLIP and DINO reaching a performance level barely above random guessing.
>
> #### ImageNet-100, Fine-Tuning on Distilled Images
> | **Train Set (1 Img/Cls)** | **CLIP**      | **DINO**      | **EVA**        | **MoCo**        | **Average**    |
> |---------------------------|---------------|---------------|----------------|-----------------|----------------|
> | **Distilled (Ours)**      | 2.7 ± 0.3     | 3.7 ± 0.2     | 12.7 ± 6.7     | **46.6 ± 12.9** | **16.4 ± 5.0** |
> | **Neighbors**             | 3.1 ± 0.2     | 4.2 ± 0.4     | 14.4 ± 9.8     | 33.5 ± 13.8     | 13.8 ± 6.0     |
> | **Centroids**             | **4.4 ± 0.3** | **4.8 ± 0.4** | 17.8 ± 11.9    | 38.0 ± 13.0     | 16.2 ± 6.4     |
> | **Random**                | 2.5 ± 0.3     | 2.6 ± 0.5     | **32.9 ± 2.5** | 21.1 ± 1.9      | 14.8 ± 1.3     |
> | **Full Dataset**          | 92.5 ± 0.0    | 95.2 ± 0.1    | 94.1 ± 0.1     | 89.4 ± 0.3      | 92.8 ± 0.1     |
>
> > What about other trajectory matching methods?
>
> Trajectory matching methods suffer from having to track gradients through an expensive inner loop.
> This issue is further exacerbated when introducing a large vision transformer as a feature extractor.
> At this point, it does not seem feasible to apply existing trajectory matching methods to the space of a pre-trained feature extractor.
>
> Fortunately, single-step gradient matching should be adequate for a linear classifier since the points along an optimal
> trajectory should be approximately co-linear.
>
> ### Overall
>
> Thank you again for taking the time to review our paper.
>
> We hope we have properly conveyed the significance of our work.
>
> Please let us know during the discussion period if there are any other points we can clarify.

---

> > ### Comment · Reviewer_gLBa · 2025-08-02
> > **agree the work is significantly different from simple application of GM**
> >
> > Thank you for the clarification on making it scale. I will raise my score.

---

> > > ### Author Response · Authors · 2025-08-07
> > >
> > > Thanks for your response!
> > >
> > > We are happy to hear that we've cleared up any points of confusion; your feedback is greatly appreciated.

---

### Official Review · Reviewer_Cicx · 2025-06-30

**Clarity:** 2
**Significance:** 2
**Originality:** 3
**Rating:** 5
**Confidence:** 3

**Summary:**

The paper describes a method, Linear Gradient Matching, that optimizes a synthetic image generation model to produce gradients similar to real data. Prototypical training images generated with this approach suffice to train capable classifiers. Several image encoders are used on top of the synthetic and real images, and they passed through randomly initialized layers to

**Questions:**

* What do you mean on line 96 with sampling a classifier at each distillation step, what is a distillation step?
* How do you generate the synthetic images? Do you use GANs, diffusion or flow based models? Do you finetune, train from scratch? How many parameters? This could be a part of figure 2
* Is the flow like this: generative model -> encoder (e.g. dino) -> learned linear projection W -> classify -> get loss -> propagate to the generative model and W
* What is the batch size used for training?

**Ethical Concerns:**

["NO or VERY MINOR ethics concerns only"]

**Final Justification:**

During my initial reading of the paper, a core part of the methodology flew over my head. After clarification from the authors, I now understand what they are doing and where my confusion came from. The added fine-grained results, error bars etc. cover the rest of my worries. I am thus happy to raise my score.

**Limitations:**

Yes

**Quality:**

2

**Strengths And Weaknesses:**

Strengths
* I find the approach intuitive, matching the learning signal, instead of surface similarity makes a lot of sense
* The paper is mostly easy to follow and well written
* Clear empirical quantitative and qualitative evaluations, e.g. the pyramid approach
* Four vision encoders are used to get loss
* Both high level iNet and spurious datasets are used

Weaknesses
* Unfortunately I do not find the presentation of the core method clear. What parameters are being updated, and what exactly is being optimized (recommend you clarify this pedagogically, this is the main issue for me to raise my score) – see questions
* Missing training details (even if the model being trained was clear to me)
* Would have been good to see an ablation with multiple images per class
* Would have been good to see how the approach works on a few fine grained datasets
* No information about how expensive the method is
* No error bars, not even bootstrapped


Notes
* I would summarize the pyramid findings when you mention figure 3 in 3.2, referencing table 2 and 3.
* The mentions of the platonic representation hypothesis seem a bit tangential and perhaps not necessary

---

> ### Author Rebuttal · Authors · 2025-07-31
>
> Thank you for your thoughtful review.
> We appreciate the constructive feedback and have already used it to improve the paper.
>
> We hope to resolve your concerns by first addressing your specific questions and then the broader concerns.
>
> ## Response to "Questions"
>
> > What do you mean on line 96 with sampling a classifier at each distillation step, what is a distillation step?
>
> We have made this clearer in the revised version of the paper, but our method can be broken up into two phases:
> the Distillation Phase and the Evaluation Phase.
>
> The "Distillation Phase" is the core of the method. This is the part where we perform the gradient matching and update the synthetic images.
>
> In the "Evaluation Phase," we evaluate the synthetic images by *training* a new linear classifier from scratch on the
> distilled data and then *evaluating* the linear classifier on the real test set.
>
> So to answer your question, a "distillation step" involves
> 1. Passing a real/synthetic batch of images through the encoder *ϕ* and a randomly initialized linear classifier *W*,
> 2. Getting the real/synthetic classification loss with respect to the real/synthetic images,
> 3. Getting the *gradients* of the real/synthetic classification losses with respect to *W*
> 4. Getting the meta loss by measuring the distance between the real and synthetic gradients
> 5. Updating the synthetic images by calculating the gradient of this meta loss with respect to the synthetic images (a bi-level optimization)
>
> > How do you generate the synthetic images? Do you use GANs, diffusion or flow based models? Do you finetune, train from scratch? How many parameters? This could be a part of figure 2
>
> We apologize for the confusion; we hope to clear up our optimization process here.
>
> To be clear, the images are ***not*** the output of a generative model.
>
> Rather, the images themselves are directly optimized using the bi-level optimization described in the previous answer.
> As such, there is no fine-tuning or training from scratch since there is no generative model.
>
> The naive approach to image optimization is to directly optimize the *pixels* of the images, but we find that this performs quite poorly due to overfitting (Table 3).
>
> Instead, we parameterize each image as a pyramid of different resolutions (Section 3.2) and reconstruct the image before each distillation step.
>
> In practice, this equates to optimizing the *Laplacian pyramid* of an image rather than its pixels directly.
> While having the same solution set as directly optimizing the images, optimizing the pyramid representation instead acts as an implicit regularizer to combat over-fitting.
> The visual difference of the naive pixel versus pyramid optimizations can be seen in Figure 3.
> Another example of such an implicit regularizer would be optimizing the images' fourier coefficients instead of the pixels directly [1], but we found the pyramid approach to work better.
>
> [1] Distill.Pub, Feature Visualization ([link](https://distill.pub/2017/feature-visualization/))
>
> > Is the flow like this: generative model -> encoder (e.g. dino) -> learned linear projection W -> classify -> get loss -> propagate to the generative model and W
>
> There are a few misconceptions here to clear up.
> 1. As discussed in the previous response, there is no generative model; the images themselves are directly optimized.
> 2. *W* is **not** learned at distillation time (only at *evaluation* time to measure the quality of our distilled images).
>
> > What is the batch size used for training?
>
> At *distillation time* (where we optimize the synthetic images), the "batch size" is equal to the number of synthetic images multiplied by the number of augmentations per batch.
> We sample an equal number of real images at each step to keep the real and synthetic batch sizes equal.
>
> At *evaluation time* (where we train a new linear classifier on the synthetic images to measure their quality), the batch size is 100.
>
> ## Response to "Weaknesses"
>
> Now we will attempt to address the broader weaknesses you mention.
>
> > Unfortunately I do not find the presentation of the core method clear. What parameters are being updated, and what exactly is being optimized
> >
> > Missing training details
>
> We hope the detailed responses in the previous section have alleviated these concerns.
>
> To briefly recap:
> * There is no generative model; we directly optimize a small set synthetic images themselves
> * The linear classifier is randomly sampled at each "distillation step" (i.e., every time we do gradient matching). It is only ever *trained* at evaluation time when we are measuring the performance of the distilled images.
>
> > Would have been good to see an ablation with multiple images per class
>
> Below is a table comparing our method to various baselines for ImageNet-100 at 10 images per class.
> We see the same trends as with 1 Image per Class overall, but the absolute gap between our method and the baselines is now smaller since we are closer to the upper bound (Full Dataset).
>
> ### ImageNet-100, 10 Img/Cls, Baseline Comparisons
>
> | Train Set (10 Img/Cls) | CLIP           | DINO           | EVA            | MoCo           | Average        |
> |------------------------|----------------|----------------|----------------|----------------|----------------|
> | **Distilled (Ours)**   | **86.1 ± 0.1** | **92.6 ± 0.1** | **90.6 ± 0.1** | **84.2 ± 0.1** | **88.4 ± 0.1** |
> | Neighbors              | 82.6 ± 0.1     | 91.0 ± 0.1     | 88.2 ± 0.2     | 83.0 ± 0.1     | 86.2 ± 0.1     |
> | Centroids              | 77.4 ± 0.2     | 87.3 ± 0.2     | 83.1 ± 0.2     | 77.8 ± 0.2     | 81.4 ± 0.2     |
> | Random                 | 85.0 ± 0.4     | 91.0 ± 0.2     | 88.0 ± 0.4     | 82.9 ± 0.4     | 86.7 ± 0.3     |
> | Full Dataset           | 92.5 ± 0.0     | 95.2 ± 0.1     | 94.1 ± 0.1     | 89.4 ± 0.3     | 92.8 ± 0.1     |
>
>
> Below is a table measuring our method's cross-model performance for ImageNet-100 at 10 images per class.
> Again, we see the same overall trends as with 1 Image per Class: DINO with the best overall generalization and a large negative outlier between MoCo and CLIP.
>
> ### ImageNet-100, 10 Img/Cls, Cross-Model Performance
>
> | Distill Model    | DINO       | EVA        | CLIP       | MoCo       | Average    |
> |------------------|------------|------------|------------|------------|------------|
> | **DINO**         | 92.6 ± 0.1 | 88.4 ± 0.1 | 85.4 ± 0.2 | 79.8 ± 0.4 | 86.6 ± 0.2 |
> | **EVA**          | 90.1 ± 0.1 | 90.6 ± 0.1 | 86.7 ± 0.1 | 76.2 ± 0.3 | 85.9 ± 0.1 |
> | **CLIP**         | 86.8 ± 0.3 | 86.2 ± 0.2 | 86.1 ± 0.1 | 66.8 ± 0.1 | 81.5 ± 0.2 |
> | **MoCo**         | 88.5 ± 0.1 | 83.7 ± 0.1 | 78.3 ± 0.1 | 84.2 ± 0.1 | 83.7 ± 0.1 |
> | **Full Dataset** | 95.2 ± 0.1 | 94.1 ± 0.1 | 92.5 ± 0.0 | 89.4 ± 0.3 | 92.8 ± 0.1 |
>
> > Would have been good to see how the approach works on a few fine grained datasets
>
> Thank you for this suggestion!
>
> We conducted experiments on CUB-200-2011 (a dataset of 200 bird species) and Stanford Dogs (a dataset of 120 dog breeds).
>
> ### CUB-200-2011
>
> | Train Set (1 Img/Cls) | CLIP           | DINO           | EVA            | MoCo           | Average        |
> |-----------------------|----------------|----------------|----------------|----------------|----------------|
> | Distilled (Ours)      | **63.2 ± 0.6** | **85.7 ± 0.2** | **75.4 ± 0.4** | **41.6 ± 0.3** | **66.5 ± 0.4** |
> | Neighbors             | 33.9 ± 0.5     | 74.9 ± 0.1     | 52.4 ± 0.4     | 21.2 ± 0.4     | 45.6 ± 0.3     |
> | Centroids             | 39.6 ± 0.3     | 76.4 ± 0.3     | 55.2 ± 0.4     | 23.8 ± 0.4     | 48.7 ± 0.4     |
> | Random                | 37.2 ± 1.4     | 62.8 ± 1.6     | 44.1 ± 1.6     | 15.2 ± 1.0     | 39.8 ± 1.4     |
> | **Full Dataset**      | 77.5 ± 0.7     | 90.2 ± 0.2     | 84.0 ± 0.3     | 43.7 ± 0.8     | 73.8 ± 0.5     |
>
> ### Stanford Dogs
>
> | Train Set (1 Img/Cls) | CLIP           | DINO           | EVA            | MoCo           | Average        |
> |-----------------------|----------------|----------------|----------------|----------------|----------------|
> | Distilled (Ours)      | **52.9 ± 0.6** | **82.4 ± 0.2** | **74.7 ± 0.4** | **68.8 ± 1.2** | **69.7 ± 0.6** |
> | Neighbors             | 21.9 ± 1.4     | 69.5 ± 0.3     | 53.2 ± 0.4     | 50.8 ± 0.8     | 48.9 ± 0.7     |
> | Centroids             | 32.4 ± 1.7     | 71.3 ± 0.4     | 51.3 ± 0.7     | 47.5 ± 1.0     | 50.7 ± 0.9     |
> | Random                | 21.0 ± 1.9     | 50.6 ± 2.1     | 36.3 ± 1.3     | 27.6 ± 2.1     | 33.8 ± 1.9     |
> | **Full Dataset**      | 76.9 ± 0.1     | 88.6 ± 0.1     | 82.6 ± 0.1     | 72.3 ± 0.5     | 80.1 ± 0.2     |
>
> The performance gap between our method and the baselines is *significantly* larger on these fine-grained datasets
> compared to ImageNet.
>
> It seems that the distilled images focus heavily on the fine-grained details necessary to distinguish between the highly similar classes.
>
> > No information about how expensive the method is
>
> In the **Compute Budget** section of the Appendix (in the included supplementary material), we note that "Distilling ImageNet-100 with the default settings takes about 4.5 hours using 4 H200 GPUs
> and around 6 hours using 4 A100 GPUs." It was difficult to get an accurate measure of how long it took to distill ImageNet-1k since we were largely bottle-necked by our network speed, but it typically took around 12 hours.
>
> > No error bars, not even bootstrapped
>
> We have since had time to repeat our experiments and now report all results as mean ± stdev over 5 runs.
>
> For the sake of space, we will not re-include all the updated tables from the paper here, but you can see examples of
> the "error bars" as ± stdev in the previous responses.
>
> ## Overall
>
> Thank you again for your constructive review.
>
> Your feedback has helped us make our paper even stronger by increasing the clarity and including more compelling results.
>
> We hope to answer any lingering questions during the upcoming discussion period.

---

> ### Comment · Reviewer_Cicx · 2025-08-01
>
> Thanks for the thorough answers. And sorry about the original review summary having been cut off, not sure what happened there. Happy to see the FGVC results turning out so nicely.
>
> I now understand what you are doing, thanks for clarifying! I will raise my score.

---

> > ### Author Response · Authors · 2025-08-07
> >
> > Thank you for your response!
> >
> > Indeed, the FGVC results turned out phenomenally and are a great addition to our paper. It is unfortunate that we cannot share visual results here, as the distilled classes look quite interesting. They are, however, included in the updated version of the paper.
> >
> > If you think of any last minute questions, we'll happily do our best to answer before the end of the discussion period.
> >
> > Thanks again!

---

> > > ### Comment · Reviewer_Cicx · 2025-08-07
> > >
> > > Maybe just a quick question on the FGVC images, do you see these as highlighting discriminatory features that can be used to interpret the target classifier? If so I can imagine one doing this to different classifiers as an interpretability thing, would be interesting to see if different classifiers use different features.

---

> > > > ### Author Response · Authors · 2025-08-07
> > > >
> > > > Yes, this is what we believe is happening!
> > > >
> > > > We have a good example of this in the initial version of the paper in Figure 7. We see that the images distilled with DINO-v2 contain (to some degree) the distinct dog breeds, while the images distilled with MoCo-v3 focus almost entirely on the spurious background correlations (forest, mountain, desert, snow). This helps explain why DINO-v2 (81.4%) performs so much better than MoCo-v3 (36.7%) when trained on the full dataset (bottom row of Table 6).
> > > >
> > > > We believe the same is true of the other datasets as well. The distilled images contain the features necessary for the classifiers to learn to distinguish between the fine-grained classes.
> > > >
> > > > In Figures B.1-B.4 of our included appendix, we include *all* the classes of ImageNet-100 distilled using the 4 base models. There are some general trends visible across the models, such as the CLIP images being more blue-green, the DINO-v2 images being more object-centric, the EVA-02 images being more green, and the MoCO-v3 images being more texture-focused. It is not yet clear exactly *why* we see these trends, but we feel that further investigation will lead to answers in model interpretability.
> > > >
> > > > Thank you for the question! Please let us know if there is anything else we can answer.

---

> > > > > ### Comment · Reviewer_Cicx · 2025-08-08
> > > > >
> > > > > Thanks, I'm in favour of acceptance, good luck!

---

### Official Review · Reviewer_pNCJ · 2025-07-03

**Clarity:** 4
**Significance:** 4
**Originality:** 4
**Rating:** 5
**Confidence:** 4

**Summary:**

The authors propose a new method for creating synthetic training data.  The method uses *Linear Gradient Matching* to create a synthetic image that has the same gradients in a linear classifier as data in the original target class.  The synthetic images successfully train different model architectures, suggesting that a similar feature representation is shared between disparate model architectures.  The method is tested using several datasets and models.

**Questions:**

1. The presented work is only valid for linear classifiers.  What assumptions for non-linear classifiers would cause the proposed method to fail?

2. Since the feature extraction backbones are pre-trained, is it necessary to use the full dataset for distillation? Would it be possible to use the linear gradients as a pre-screening method to identify a smaller subset of data which is capable of achieving the same results without the extensive compute resources?

3. Please provide pseudo code for the main training algorithms until the main code repository is suitable for release

4. Would it be possible to provide a toy example that allows users to apply the technique to smaller datasets?

**Ethical Concerns:**

["NO or VERY MINOR ethics concerns only"]

**Final Justification:**

The authors clarified my concerns regarding compute and the generalizability of this work. Including the pseudocode makes the work accessible to a broader audience, and I hope that the authors will release the full code repository on acceptance. If this work is broadly implemented across different fields, it could provide good experimental confirmation (or refutation) for different theories in foundations of AI, including universal feature approximations and desirable properties for model hessian spectra.

**Limitations:**

Because of the extensive compute and data required for this paper, it will be challenging for the community to apply this to smaller datasets or verify the results independently.

**Quality:**

4

**Strengths And Weaknesses:**

**Strengths**

I think this paper does a good job providing a novel experimental method that touches on SOTA models, interesting theoretical underpinnings, and a practical problem.   The presented experiments are convincing, extensive, and provide a good direction for future work.

**Weaknesses**

The code is currently unavailable, although the authors state their intention to release code upon acceptance.  The issue with this is that the authors report that the proposed method's success largely relies on the pyramid upsampling method, but details of the pyramid construction ("we progressively optimize our pyramid, starting with just the lowest-resolution component and periodically adding more tiers during distillation...") are not presented in enough detail to re-create the code from the paper alone, or to understand how this optimization affects the data distillation method proposed.  Although this is discussed further in the supplementary info, a complete algorithm or pseudo code is omitted from the paper. It also isn't clear why the pyramid plays such a large role if the linear gradient matching is the primary theoretical explanation for why the method works.

---

> ### Author Rebuttal · Authors · 2025-07-31
>
> Thank you for your valuable feedback;
> you have helped us identify several areas of improvement which have already further improved the quality of our paper.
>
> We will now address the concerns and questions raised in your review.
>
> ## Addressing "Questions"
>
> > The presented work is only valid for linear classifiers. What assumptions for non-linear classifiers would cause the proposed method to fail?
>
> We chose to study linear classifiers in this work simply because they (along with fine-tuning) are the standard for measuring the quality of a pre-trained model.
> Our method may indeed transfer to non-linear classifiers, but we did not have time to adequately explore this direction during the rebuttal week.
> While single-step gradient matching is adequate for a linear model, a non-linear classifier might require adapting trajectory-matching techniques from previous dataset distillation methods.
> This would introduce memory issues due to the large footprint of the pre-trained feature extractor.
> We intend to investigate this further in future work.
>
> On the other hand, a full-scale fine-tuning would *certainly* prove problematic.
> When distilling a dataset down to a single image per class, fine-tuning the entire large pre-trained feature extractor
> tends to lead to catastrophic overfitting.
> We present a table here for ImageNet-100 when fine-tuning on the distilled data.
>
> | Train Set (1 Img/Cls) | CLIP          | DINO          | EVA            | MoCo            | Average        |
> |-----------------------|---------------|---------------|----------------|-----------------|----------------|
> | **Distilled (Ours)**  | 2.7 ± 0.3     | 3.7 ± 0.2     | 12.7 ± 6.7     | **46.6 ± 12.9** | **16.4 ± 5.0** |
> | Neighbors             | 3.1 ± 0.2     | 4.2 ± 0.4     | 14.4 ± 9.8     | 33.5 ± 13.8     | 13.8 ± 6.0     |
> | Centroids             | **4.4 ± 0.3** | **4.8 ± 0.4** | 17.8 ± 11.9    | 38.0 ± 13.0     | 16.2 ± 6.4     |
> | Random                | 2.5 ± 0.3     | 2.6 ± 0.5     | **32.9 ± 2.5** | 21.1 ± 1.9      | 14.8 ± 1.3     |
> | Full Dataset          | 92.5 ± 0.0    | 95.2 ± 0.1    | 94.1 ± 0.1     | 89.4 ± 0.3      | 92.8 ± 0.1     |
>
> > Since the feature extraction backbones are pre-trained, is it necessary to use the full dataset for distillation? Would it be possible to use the linear gradients as a pre-screening method to identify a smaller subset of data which is capable of achieving the same results without the extensive compute resources?
>
> The compute required for our method scales with the *number of classes* in the real dataset, **not** the number of samples.
> Currently, the batch size is equal to the number of images per class (typically 1) times the
> augmentations per batch (typically 10) times the number of classes.
>
> So in short, pruning the real dataset before distillation will not reduce the method's cost, but future work may lead to
> methods that can get away with smaller batch sizes.
>
> > Please provide pseudo code for the main training algorithms until the main code repository is suitable for release
>
> Of course, please see below.
>
> ```python
> import torch
> from typing import List
> def reconstruct_pyramid(pyramid: List[torch.Tensor]) -> torch.Tensor:
>     # The pyramid is stored as a list of Tensors of different shapes
>     # (3, 1, 1), (3, 2, 2), (3, 4, 4), (3, 8, 8), (3, 256, 256)
>     sum = torch.zeros(3, 256, 256)
>     for layer in pyramid:
>         # each layer is bilinearly up-sampled to the final resolution and added together
>         sum += F.interpolate(layer, (3, 256, 256), "bilinear")
>     # we normalize by the current number of layers to account for progressively adding more layers
>     image = sum / len(pyramid)
>     return torch.tanh(image)
> ```
>
> Above is the code for reconstructing the distilled image from its pyramid representation.
> In practice, this is equivalent to reconstructing an image from its Laplacian pyramid.
> At initialization, the pyramid only consists of the 1x1 layer.
> An additional layer is added every 200 distillation steps until the final layer (256x256) has been added.
>
> ```python
> import torch
> def distillation_step(
>         pyramids: List[List[Tensor]]# our set of distilled image pyramids
>         labels: Tensor              # labels for distilled images/pyramids
>         optim: torch.optim.Adam,    # optimizer for distilled pyramids
>         backbone: nn.Module,        # the pre-trained feature extractor
>         num_features: int,          # number of features in the backbone's output
>         num_classes: int,           # number of classes in the dataset
>         augs_per_batch: int,        # how many sets of augmentations per batch (typically 10)
>         augment: Callable,          # augmentation function
> ):
>     # initialize a random linear classifier
>     W = torch.randn(num_features, num_classes)
>
>     # get synthetic images by reconstructing from their pyramids
>     x_syn = [reconstruct_pyramid(p) for p in pyramids]
>     y_syn = labels
>     # we augment the synthetic images multiple times to get better gradients
>     x_syn = torch.cat([augment(x_syn) for i in range(augs_per_batch)])
>
>     z_syn = backbone(x_syn)
>     out_syn = torch.matmul(z_syn, W)
>     loss_syn = cross_entropy(out_syn, y_syn)
>
>     # we have to retain the graph here because we will backprop through this backprop later
>     grad_syn = torch.autograd.grad(loss_syn, W, retain_graph=True, create_graph=True).flatten()
>
>     # load from disc, this is our biggest bottleneck because we have to load over a network filesystem
>     x_real, y_real = get_real_data()
>     x_real = augment(x_real)
>
>     z_real = backbone(x_real)
>     out_real = torch.matmul(z_real, W)
>     loss_real = cross_entropy(out_real, y_real)
>
>     grad_real = torch.autograd.grad(loss_real, W).flatten()
>
>     # our overall loss is the distance between the real and synthetic gradients w.r.t. W
>     meta_loss = 1 - cosine_similarity(grad_syn, grad_real)
>
>     optim.zero_grad()
>     # the meta loss is back-propagated through the earlier synthetic gradient calculation
>     # it is a "bi-level" optimization
>     # the loss is back-propagated past the synthetic images all the way to their pyramid representations
>     meta_loss.backward()
>     optim.step()
> ```
>
> Above is the pseudo code for a single distillation step.
> At each step, we:
> 1. Sample a random linear classifier
> 2. Reconstruct the synthetic images from their pyramids and augment them.
> 3. Get the gradient of the classification loss w.r.t. W using the *synthetic* images.
> 4. Load real images from disc.
> 5. Get the gradient of the classification loss w.r.t. W using the *real* images.
> 6. Calculate the meta loss as the distance between these two gradients.
> 7. Back-propagate the meta loss through the initial gradient calculation all the way to the distilled pyramids.
>
> > Would it be possible to provide a toy example that allows users to apply the technique to smaller datasets?
>
> Yes! In fact, we already included some results several such datasets.
>
> As discussed earlier, the compute requirements scale with the number of *classes* in the dataset, so any dataset with
> fewer classes will distill much faster.
>
> The 4-class (Spawrious) and 2-class (Waterbirds) datasets discussed in Section 4.6 were distilled in under 20 minutes on a single GPU.
>
> The 10-class ArtBench dataset from section 4.7 was distilled in 30 minutes on a single GPU.
>
> ## Addressing "Weaknesses"
> > The code is currently unavailable ...
>
> Thank you for bringing to our attention how this can be challenging for reviewers;
> we will strive to include clean code bases in future submissions.
>
> We hope that the pseudo-code included in the previous responses adequately answer all your questions about our method.
>
> > It also isn't clear why the pyramid plays such a large role if the linear gradient matching is the primary theoretical explanation for why the method works.
>
> The pyramid representation solely acts as an implicit regularizer on the distilled images while still allowing for the
> same potential set of solutions.
>
> An example of another such implicit regularizer would be optimizing the fourier coefficients of the images instead of the pixels directly.
> While still allowing for the same set of result images, this alternate representation affects the direction of steepest
> descent [1], thereby affecting the optimization dynamics and resulting in fewer high-frequency artifacts.
>
> We simply found that optimizing the pyramid representation as an implicit regularizer yielded better results than other
> methods, such as optimizing the fourier coefficients.
> Furthermore, we found that progressively adding more layers to the pyramid over the course of distillation performed
> better than initializing it with all layers already in-tact.
> Qualitatively, we observed that this progressive method led to more global patterns across the images.
>
> [1] Distill.Pub, Feature Visualization ([link](https://distill.pub/2017/feature-visualization/))
>
> ## Addressing "Limitations"
> > Because of the extensive compute and data required for this paper, it will be challenging for the community to apply this to smaller datasets or verify the results independently.
>
> As discussed in our earlier responses, the compute requirements for our method scale solely with the number of classes
> in the dataset.
> As such, only the full ImageNet-1k was prohibitively expensive since the batch size was so large.
>
> In fact, as noted in the "Compute Budget" section of our included supplementary material,
> it takes 6 hours to distill ImageNet-100 using 4 A100 GPUs or 4.5 hours using 4 H200 GPUs.
>
> Depending on a user's compute budget, this may still be considered expensive.
> However, distilling a 10-class dataset like ArtBench or ImageWoof takes only 40 minutes on a single RTX4090 GPU.
> This should fall comfortably within most users' compute budgets if ImageNet-100 distillation is too expensive.

---

> > ### Comment · Reviewer_pNCJ · 2025-08-06
> >
> > Thank you to the authors for addressing my concerns; I appreciate their responses. I will raise my score for the quality of the work.

---

> > > ### Author Response · Authors · 2025-08-07
> > >
> > > Thank you for your response!
> > >
> > > We are glad to hear we've cleared up any points of confusion. Your feedback has helped us improve our paper by increasing the clarity of our methods sections, making it easier for readers to absorb.
> > >
> > > If you think of any last-minute questions, we'll happily do our best to answer by the end of the discussion period.
> > >
> > > Thanks again!

---

> ### Comment · Area_Chair_7Ymd · 2025-08-05
> **post-rebuttal comments**
>
> Dear reviewer pNCJ,
>
> As you may have seen already, the authors have responded to the questions in your initial review. Can you please share your thoughts post rebuttal, once you've had a chance to see the author responses and also the other reviews?
>
> Best, AC

---

### Official Review · Reviewer_Kyf1 · 2025-07-05

**Clarity:** 4
**Significance:** 2
**Originality:** 3
**Rating:** 4
**Confidence:** 4

**Summary:**

The paper tackles the problem of dataset distillation for linear classifiers built on top of fixed, pre-trained vision encoders. It proposes a method that optimizes synthetic images such that, when passed through the frozen feature extractor, they induce gradients that closely match those produced by real training data. The approach is evaluated across a range of popular vision encoders, including MoCo v3, DINO v2, EVA v2 and CLIP. Experimental results show that the proposed technique outperforms baselines of nearest neighboors, random images and centroids on several benchmarks. The paper also investigates its application to model interpretability, suggesting that the method offers insights into the relationship between input images and classifier behavior.

**Questions:**

I found the experiments in Table 6 particularly interesting. While the models achieve similar performance on ImageNet, there is a significant gap between DINO v2 and MoCo v3 on the spurious background dataset. If I understand correctly, all "DINO" models in the paper refer to DINO-v2. (A minor suggestion is to consistently use “DINOv2” throughout the paper to avoid confusion.) Given that both DINOv2 and MoCo v3 are based on contrastive learning, this performance gap raises intriguing questions. One possible explanation is the use of a patch-level loss in DINOv2, which might make it more robust to spurious correlations. Alternatively, it could be related to the use of smaller crops in DINOv1. Including experiments on DINOv1 could help disentangle these factors and provide further insight into the source of robustness.

**Ethical Concerns:**

["NO or VERY MINOR ethics concerns only"]

**Final Justification:**

My questions are properly addressed by the rebuttal. I have raised my confidence score accordingly.

**Limitations:**

Yes

**Paper Formatting Concerns:**

No.

**Quality:**

3

**Strengths And Weaknesses:**

Strengths:

- The paper is well-written and clearly presented.

- It conducts thorough ablations of key components in the proposed technique, including Pyramid Representations, and Differentiable Augmentations, showing that both are crucial for achieving strong performance.

- The method is evaluated across a variety of pre-trained vision encoders and includes insightful analyses such as:

  - Cross-model performance comparisons,

  - Evidence supporting the platonic representation hypothesis,

  - Identifying spurious correlations and analyzing out-of-distribution behavior.

Weaknesses:

- The motivation for focusing specifically on dataset distillation for linear classifiers is underdeveloped:

  - Why is this particular task important?

  - Why prioritize linear probing over more general settings like full model fine-tuning?

- While the dataset distillation technique is well-executed, the paper might have been even stronger if it had been framed primarily as a study in model interpretability, using dataset distillation as a tool rather than the main focus. In this framing, the cross-model comparisons and analyses could have been pushed further to explore deeper insights into the behavior and differences among pre-trained models.

---

> ### Author Rebuttal · Authors · 2025-07-31
>
> Thank you for your constructive comments.
> The insights provided have already improved our paper and inspired us to continue pushing along this line of research.
>
> We hope to resolve your concerns by first addressing your broader points followed by the specific questions.
>
> ## Addressing "Weaknesses"
> >The motivation for focusing specifically on dataset distillation for linear classifiers is underdeveloped:
> >Why is this particular task important?
>
> The traditional field of dataset distillation has somewhat stagnated;
> while promising results have been seen on toy datasets like CIFAR, achieving even moderate success on more realistic
> datasets like ImageNet requires so many synthetic images per class that it almost goes against the spirit of the task.
> For ImageNet-1k, from-scratch dataset distillation methods have not come close to 5% accuracy at 1 image per class,
> severely limiting the distilled data's usefulness.
>
> Given that most vision tasks are solved today by starting from a pre-trained feature extractor rather than training a
> model from scratch, we feel it is time for dataset distillation to explore this area as well.
>
> While prior distillation methods worked well in small-scale settings, none of them scaled up to large models or
> realistic datasets (without distilling a very large number of synthetic images per class), calling into question the field's practical usefulness.
> Using our method, it is possible to train a linear classifier to nearly **75%** test accuracy on ImageNet-1k with just a
> single image per class in under a minute. Since most models used today start from a pre-trained backbone, the data
> distilled with our method show potential for practical use cases rather than just toy problems.
>
> > Why prioritize linear probing over more general settings like full model fine-tuning?
>
> Thank you for raising this question, as the answer is not immediately intuitive.
>
> When distilling a dataset down to just one (or a handful of) image(s) per class, fine-tuning an entire large vision
> model on such a small set of images tends to lead to catastrophic overfitting.
>
> Overfitting to the distilled dataset is a known issue of dataset distillation, but training a model of such high
> capacity on the distilled dataset dramatically exacerbates the issue.
>
> Below, we include a table of results obtained by fine-tuning the models on the distilled data from ImageNet-100.
> We observe the expected trend: fine-tuning the models on the tiny distilled datasets leads to catastrophic overfitting.
>
> We will touch on this briefly in the main paper and include the quantitative results in the appendix.
>
> #### ImageNet-100, Fine-Tuning on Distilled Images
> | **Train Set (1 Img/Cls)** | **CLIP**      | **DINO**      | **EVA**        | **MoCo**        | **Average**    |
> |---------------------------|---------------|---------------|----------------|-----------------|----------------|
> | **Distilled (Ours)**      | 2.7 ± 0.3     | 3.7 ± 0.2     | 12.7 ± 6.7     | **46.6 ± 12.9** | **16.4 ± 5.0** |
> | **Neighbors**             | 3.1 ± 0.2     | 4.2 ± 0.4     | 14.4 ± 9.8     | 33.5 ± 13.8     | 13.8 ± 6.0     |
> | **Centroids**             | **4.4 ± 0.3** | **4.8 ± 0.4** | 17.8 ± 11.9    | 38.0 ± 13.0     | 16.2 ± 6.4     |
> | **Random**                | 2.5 ± 0.3     | 2.6 ± 0.5     | **32.9 ± 2.5** | 21.1 ± 1.9      | 14.8 ± 1.3     |
> | **Full Dataset**          | 92.5 ± 0.0    | 95.2 ± 0.1    | 94.1 ± 0.1     | 89.4 ± 0.3      | 92.8 ± 0.1     |
>
> > ... the paper might have been even stronger if it had been framed primarily as a study in model interpretability, using dataset distillation as a tool rather than the main focus.
>
> Thank you for this suggestion!
>
> Indeed, we also find the interpretability results to be some of the most exciting in the paper.
> We have reworded the results section to further highlight these results and emphasize the distillation method's usefulness
> as an interpretability tool.
>
> We will more aggressively pursue this angle in future works now that the problem (Dataset Distillation for Pre-Trained
> Self-Supervised Vision Models) has been formally introduced, opening up space for ancillary endeavors.
>
> ## Addressing "Questions"
>
> > If I understand correctly, all "DINO" models in the paper refer to DINO-v2. (A minor suggestion is to consistently use “DINOv2” throughout the paper to avoid confusion.)
>
> Yes, this is true throughout the paper *except* for the section on ArtBench.
> We used DINO-v1 for ArtBench because we wanted a model trained only on ImageNet such that the artworks would be out of distribution.
>
> We initially dropped the "v2" in the tables to save space, but we see how this could cause confusion and have since labeled them more specifically as you suggested.
>
> > One possible explanation [for DINO-v2's much better performance than MoCo-v3 on the spurious background dataset] is the use of a patch-level loss in DINOv2, which might make it more robust to spurious correlations. Alternatively, it could be related to the use of smaller crops in DINOv1. Including experiments on DINOv1 could help disentangle these factors and provide further insight into the source of robustness.
>
> This is an interesting insight; thank you for the suggestion!
>
> Below, we include an extended version of the Spawrious table from the paper that includes DINO-v1 as well.
>
> #### Spawrious, DINO Comparison
>
> | Train Set (1 Img/Cls) | DINO-v1 ViT-B/16 | DINO-v2 ViT-B/14 | MoCo-v3 ViT-B/16 | Average        |
> |-----------------------|------------------|------------------|------------------|----------------|
> | Distilled (Ours)      | 48.6 ± 1.8       | **83.3 ± 3.8**   | **35.6 ± 2.3**   | **55.8 ± 2.6** |
> | Neighbors             | **50.4 ± 1.8**   | 81.7 ± 3.1       | 33.9 ± 2.5       | 55.3 ± 2.5     |
> | Centroids             | 50.0 ± 2.4       | 80.7 ± 2.8       | 31.3 ± 1.9       | 54.0 ± 2.4     |
> | Random                | 46.7 ± 3.4       | 64.2 ± 3.5       | 33.5 ± 8.4       | 48.1 ± 5.1     |
>
> This DINO-v1 model has a similar patch size as the DINO-v2 model (16 versus 14).
>
> We see that this DINO-v1 model, like MoCo-v3, also performs significantly worse than DINO-v2 on this spurious background dataset.
>
> Qualitatively, we observe that the images distilled using DINO-v1 also seem to focus too much on the background environments (although we have no way of sharing these images in the rebuttal).
>
> However, we still cannot be sure of the precise reason for this performance gap.
> It could be due to DINO-v2's patch-based losses, but it could also simply be due to the training data.
> DINO-v1 and MoCo-v3 were both trained on ImageNet, while DINO-v2 was trained on an undisclosed highly-curated dataset.
>
> Regardless, it is quite clear that this distillation method can indeed act as a useful tool with which to study model interpretability.
> We plan on further exploring this direction in future work.
>
> ## Overall
>
> Thank you again for your thoughtful review.
>
> We appreciate your enthusiasm for the interpretability results, and we aim to explore this direction more deeply in future work.
>
> We hope to answer any lingering questions during the upcoming discussion period.

---

> > ### Comment · Reviewer_Kyf1 · 2025-08-06
> >
> > Thanks a lot for the insightful feedback.
> >
> > I appreciate the explanation on the motivation and the additional results on DINO-v1 vs DINO-v2. Indeed I didn't include the vast training data difference between DINO v1 and v2. But the large gap between the two models are very interesting. I'd love to follow your future work!

---

> > > ### Author Response · Authors · 2025-08-07
> > >
> > > Thank you for your response! Indeed, the difference in training data makes it hard to discern where the differences in performance come from.
> > >
> > > We do plan to continue this line of work; in the future, we would like to retrain the base models (DINOv1/v2, MoCo, etc) on the same dataset to help disentangle the causes of performance gaps.
> > >
> > > If you think of any last-minute questions, we'll happily do our best to answer before the end of the discussion period.
> > >
> > > Thanks again!

---

> ### Comment · Area_Chair_7Ymd · 2025-08-05
> **post-rebuttal comments**
>
> Dear reviewer Kyf1,
>
> Can you please share your thoughts post rebuttal, once you've had a chance to see the author responses and also the other reviews?
>
> Best,
> AC

---

### Decision · Program_Chairs · 2025-09-17

**Decision:**

Accept (poster)

**Comment:**

This paper on dataset distillation tackles the case of pre-trained self-supervised models that are often used in computer vision problems. The core of this work is an approach for "generating" synthetic data through linear gradient matching that ensures same gradients in a linear classifier for synthetic and real data. The proposed approach is widely evaluated with several popular encoders (e.g., DINOv2, CLIP, MoCo v3) and compared against several relevant methods and baselines.

During the review process several strengths identified for this work include: a diverse set of convincing evaluations and ablation studies, presenting a relevant theoretical basis, the use of relevant datasets showing the practicality of the approach and its potential use in the research community. The initial reviews also noted questions on: the importance of this task, the motivation for using linear probing (e.g., compared to fine-tuning), reproducibility of this work, use beyond linear classifiers, the novelty of the work. The rebuttal and the discussion that ensued clarified all the questions with convincing evidence.

Thus, all the reviewers and the AC recommend the paper for acceptance. The final version of the manuscript should include the clarifications provided in the rebuttal and discussion that ensued.